# ROBUST ONION: PEELING OPEN VOCAB OBJECT DETECTORS UNDER NOISE

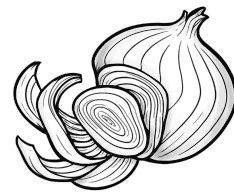

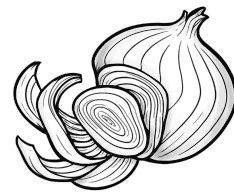

Figure 1: **Effect of Noise Degradation:** GLIP (above, 2022) & MM-GIDNO (bottom, 2024b) performance on COCO (Lin et al., 2015) for noises like turbulence, pixelation, and motion blur.

## ABSTRACT

The impact of real-world noise on **O**pen **V**ocabulary **O**bject **D**etectors (OV-ODs) is constrained by their architectural complexity and the scarcity of noise-annotated datasets. Our empirical analysis, **Robust Onion**, uses controlled synthetic visual degradations to mirror *feature collapse* of real-world noises and systematically peel apart OV-OD components to assess their robustness. Our findings include: Similar vision backbones show comparable robustness, driven by identical feature collapse at similar layers. Pretraining, architectural nuances, and captions contribute little to robustness. Robustness relies strongly on the image domain rather than on annotations, explaining the similar impact of COCO and LVIS on robustness (same images, different annotations), and how datasets like ODinW-13, with large, isolated objects, can give a misleading impression of high robustness. These insights point to potential research on cross-layer feature exchange and continual learning strategies to improve robustness efficiently. Our findings highlight critical directions for designing robust OV-ODs under challenging visual degradations.

## 1 INTRODUCTION

Vision Language Models (VLMs) have shown strong generalization in tasks like image-to-text retrieval (Saha et al., 2024), open-vocabulary classification (Abdelhamed et al., 2025), image captioning (Cheng et al., 2025), and visual-question-answering (Huynh et al., 2025). The ability to adapt without fine-tuning makes VLMs highly beneficial for applications where *zero-shot* is not just a convenience but a necessity. VLM based **Open Vocabulary Object Detectors (OV-ODs)** are rapidly gaining attention for their utility and advantages in security (He et al., 2024), medical imaging (Yu et al., 2025), environmental monitoring (Xue et al., 2024), and self-driving cars (Tian et al., 2024).

Real-world deployment of OV-ODs requires a critical understanding of their robustness against visual distortions / noise. OV-ODs are among the most complex deep learning models with many moving components, such as vision-text backbones, fusion network, box predictors, alignment networks *etc*. However, analyzing real-world low-quality (LQ) and out-of-distribution noise is difficult

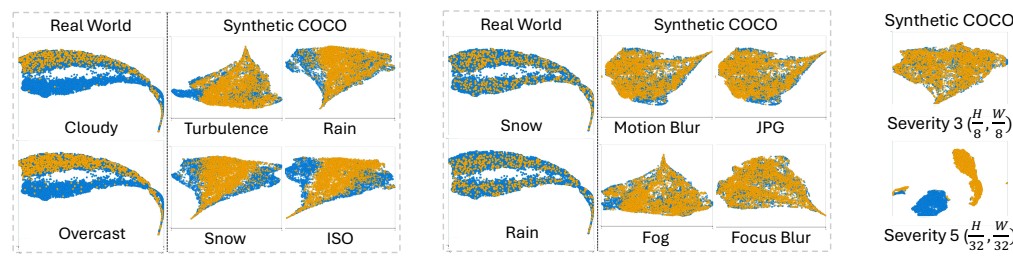

Figure 2: **Synthetic COCO & Real-World BDDK100:** GLIP-T synthetic noisy features collapse (against clean image) *aligns* with real-world collapse, tuned such that they look 'realistic' (fig. 1) for HQ-LQ pairs. All BDDK100 categories (noisy) in blue except highlighted (clean *unavailable*).

because **matching high-quality (HQ) images counterparts to LQ images are scarce**, making it almost *impossible* to isolate noise effects and build *well-annotated* noise datasets. Hence, despite their prevalence, the impact of distortions on VLMs-based OV-ODs remains largely unexplored.

Addressing the gap in robustness against real-world distortions (*e.g.* BDDK-100 (2020)), our novel analysis, **Robust Onion** broadly categorizes **noise-induced feature (variance) collapse** (Ling et al., 2023; Chai et al., 2023) into two categories: **Observable** (fig. 2a): noisy features form distinct clusters separate from clean HQ features (*e.g.* cloudy, overcast), **Minimal** (fig. 2b): little / no observable collapse (*e.g.* snow, rain). By carefully tuning synthetic noises, we mimic these feature collapses, providing a *practical proxy* for real-world degradations (*e.g.* turbulence for *observable*, motion blur for *minimal*). Increasing noise severity can change the collapse from *minimal* to *observable* (fig. 2c). Robust Onion then empirically *peels* each component of OV-OD under these controlled noises.

To answer: *"How do visual distortions impact complex models such as OV-ODs, and what are the most effective directions to improve their robustness?"*, our analysis frames four key questions: **(1)** Among all bells and whistles (*e.g.* pretraining, fine-tuning & architectural modules), what limits robustness? **(2)** Are larger models inherently more robust, or are other factors decisive? **(3)** Is robustness solely determined by the model, or do input images play a role? **(4)** Is it possible to leverage language to improve robustness under visual noise? Our analysis pinpoints the key bottlenecks and highlights why some noisy images may be easier to detect than others.

**Robust Onion** analysis of OV-ODs against visual distortions: *Vision Backbone drives robustness:* Similar backbones exhibit comparable robustness due to resembling feature collapses at similar depths. For backbones of sufficient scales (*e.g.* Swin-B), depth or size adds little to robustness, regardless of bells and whistles of pretraining or overall architecture. *Layer-wise robustness:* Shallow layers are more adversely affected by noise; cross-layer information exchange can potentially improve robustness. *Dataset bias:* ODinW-13 can give a false impression of robustness because it features large and isolated objects. *Domain over annotation:* Image domain (detecting *'on'* what) matters far more than annotation type (detecting what), explaining similar robustness of COCO & LVIS. *Minimal language influence:* Once the visual features are degraded, language/captions contribute little to recover lost robustness. Each analysis includes **[Takeaways & Model Design]** highlighting key insights for designing robust OV-OD. We conclude by advocating for a cost-efficient continual learning strategy catered to OV-ODs for improving robustness in a zero-shot setting.

## 2 RELATED WORK

**VLMs and OV-ODs:** Advances in vision-language pre-training (Radford et al., 2021; Minderer et al., 2022) has led to transferring knowledge from VLMs to object detectors (Shen et al., 2024; Gu et al., 2021; Zareian et al., 2021). Works like Alayrac et al. (2022); Tsimpoukelli et al. (2021); Chen et al. (2022); Minderer et al. (2023); Zhao et al. (2024a) have shown strong object detection capabilities through large-scale multimodal training. Versatile generalization of OV-ODs (Li et al., 2025; Deng et al., 2024) makes them ideal for real-world applications, thereby understanding their limitations (Bianchi et al., 2024; Zhang et al., 2024a) against real-world noises is important.

**Robustness against Noise:** Weather (rain, fog, turbulence) or artifacts (compression) pose significant challenges in object detection (Mao et al., 2023; Qin et al., 2022; Chhipa et al., 2024;

Zhang et al., 2024b; Yoo et al., 2024). Distortions cause loss of discriminative features, a fundamental problem affecting all models (Shermeyer & Van Etten, 2019). However, despite the prevalence of such visual noises(*e.g.* surveillance (Davila et al., 2023), satellite imagery (Patil et al., 2017), autonomous-driving (Tian et al., 2024), *etc*), their impact on VLMs remains largely unexplored (Cheng et al., 2019; Li et al., 2019). Existing works use synthetic noises to show improvement on real-world datasets like low-resolution person re-id (Pathak & Rawat, 2025), driving in fog/rain (Gupta et al., 2024b). Recently, LR0.FM (Pathak et al., 2025) benchmarked VLMs robustness for image classification. On the contrary, object detection is far more complex with significant practical use. We present a comprehensive analysis of object detection via SOTA OV-OD models, revealing bottlenecks and critical factors in their robustness.

# 3 ANALYSIS SETUP

**Models**: We analyze 6 publicly available OV-ODs: RegionCLIP (RC, RCx4, 2022), GLIP (2022), FIBER (2022), MM-Grounding-DINO (MM-GDINO, 2024b), GLEE (2024), and YOLO-World (YOLO, 2024). Figure 3 shows robustness of all models (backbones, fine-tuned & zero-shot). CNN-based ones (YOLO & RegionCLIP) are not as robust as transformer ones, hence, not the main focus of our analysis. More in *Supplementary*.

**Datasets:** We evaluate robustness on 3 benchmarks: COCO (val2017, 2015), LVIS (miniVal, 2019) and ODinW-13 (set of 13 datasets, 2022). COCO (80 categories) and LVIS (1,203 categories) have same images, but different annotations. For language, we use RefCOCO (2014), RefCOCO+ (2016), RefCOCOg (2016), and Flickr30k (2015). *Real-world* Wider Face (Yang et al., 2016), naturally occurring noisy images, is used to test our proposed solution.

**Framework:** Figure 4 illustrates a general framework for object detection. Input image and captions are processed through a pyramidal (multi-scale) vision encoder (*e.g.* ResNet, Swin Transformer) and a text encoder (*e.g.* BERT (2019), CLIP (2021), *etc.*), respectively. Vision features are enhanced via FPN (or pixel-decoder (GLEE)) followed by cross-self-attention for fusion with text embeddings. These fused features are used to predict bounding boxes, confidence scores, and class labels.

**Noises:** Analyzing noise requires measuring the drop in performance *relative* to clean features *i.e.* analysis of LQ-HQ input image pairs (rarely present in the real world). Instead, we use the two categories of feature collapse (fig. 2) to pick two controlled synthetic noises: turbulence for *observable*, and motion blur for *minimal*. We also analyze pixelation across severity (intensity). Pixelation (*e.g.* compression, distant objects) is simulated via bicubic interpolation: downsampling the image $(H, W)$ to $(\frac{H}{2^3}, \frac{W}{2^3})$ and upscaling it back (Severity 3)[1]. Tur-

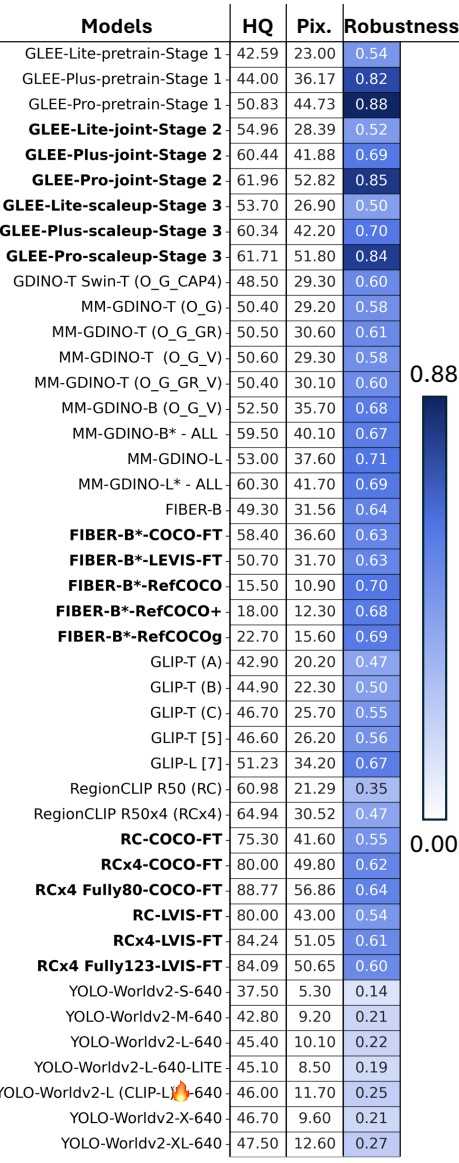

| Models | HQ | Pix. | Robustness |
|---|---|---|---|
| GLEE-Lite-pretrain-Stage 1 | 42.59 | 23.00 | 0.54 |
| GLEE-Plus-pretrain-Stage 1 | 44.00 | 36.17 | 0.82 |
| GLEE-Pro-pretrain-Stage 1 | 50.83 | 44.73 | 0.88 |
| **GLEE-Lite-joint-Stage 2** | 54.96 | 28.39 | 0.52 |
| **GLEE-Plus-joint-Stage 2** | 60.44 | 41.88 | 0.69 |
| **GLEE-Pro-joint-Stage 2** | 61.96 | 52.82 | 0.85 |
| **GLEE-Lite-scaleup-Stage 3** | 53.70 | 26.90 | 0.50 |
| **GLEE-Plus-scaleup-Stage 3** | 60.34 | 42.20 | 0.70 |
| **GLEE-Pro-scaleup-Stage 3** | 61.71 | 51.80 | 0.84 |
| GDINO-T Swin-T (O_G_CAP4) | 48.50 | 29.30 | 0.60 |
| MM-GDINO-T (O_G) | 50.40 | 29.20 | 0.58 |
| MM-GDINO-T (O_G_GR) | 50.50 | 30.60 | 0.61 |
| MM-GDINO-T (O_G_V) | 50.60 | 29.30 | 0.58 |
| MM-GDINO-T (O_G_GR_V) | 50.40 | 30.10 | 0.60 |
| MM-GDINO-B (O_G_V) | 52.50 | 35.70 | 0.68 |
| MM-GDINO-B* - ALL | 59.50 | 40.10 | 0.67 |
| MM-GDINO-L | 53.00 | 37.60 | 0.71 |
| MM-GDINO-L* - ALL | 60.30 | 41.70 | 0.69 |
| FIBER-B | 49.30 | 31.56 | 0.64 |
| **FIBER-B*-COCO-FT** | 58.40 | 36.60 | 0.63 |
| **FIBER-B*-LEVIS-FT** | 50.70 | 31.70 | 0.63 |
| **FIBER-B*-RefCOCO** | 15.50 | 10.90 | 0.70 |
| **FIBER-B*-RefCOCO+** | 18.00 | 12.30 | 0.68 |
| **FIBER-B*-RefCOCOg** | 22.70 | 15.60 | 0.69 |
| GLIP-T (A) | 42.90 | 20.20 | 0.47 |
| GLIP-T (B) | 44.90 | 22.30 | 0.50 |
| GLIP-T (C) | 46.70 | 25.70 | 0.55 |
| GLIP-T [5] | 46.60 | 26.20 | 0.56 |
| GLIP-L [7] | 51.23 | 34.20 | 0.67 |
| RegionCLIP R50 (RC) | 60.98 | 21.29 | 0.35 |
| RegionCLIP R50x4 (RCx4) | 64.94 | 30.52 | 0.47 |
| **RC-COCO-FT** | 75.30 | 41.60 | 0.55 |
| **RCx4-COCO-FT** | 80.00 | 49.80 | 0.62 |
| **RCx4 Fully80-COCO-FT** | 88.77 | 56.86 | 0.64 |
| **RC-LVIS-FT** | 80.00 | 43.00 | 0.54 |
| **RCx4-LVIS-FT** | 84.24 | 51.05 | 0.61 |
| **RCx4 Fully123-LVIS-FT** | 84.09 | 50.65 | 0.60 |
| YOLO-Worldv2-S-640 | 37.50 | 5.30 | 0.14 |
| YOLO-Worldv2-M-640 | 42.80 | 9.20 | 0.21 |
| YOLO-Worldv2-L-640 | 45.40 | 10.10 | 0.22 |
| YOLO-Worldv2-L-640-LITE | 45.10 | 8.50 | 0.19 |
| YOLO-Worldv2-L (CLIP-L🔥-640 | 46.00 | 11.70 | 0.25 |
| YOLO-Worldv2-X-640 | 46.70 | 9.60 | 0.21 |
| YOLO-Worldv2-XL-640 | 47.50 | 12.60 | 0.27 |

0.88

0.00

Figure 3: **All models on COCO (mAP):** Shade $\propto$ robustness against pixelation. Fine-tuned (COCO, LVIS, RefCOCO) in **bold**.

bulence (*e.g.* air (hot) movement) is simulated via pre-trained neural network (Mao et al., 2021). Motion Blur (Gupta et al., 2024a) simulates motion (*e.g.* videos). These noises are only applied to

---

[1]Unlike low-res images, downsampled images can have high resolution *e.g.* $\frac{H}{8} = 256$; Severity 's': $\frac{H}{2^s}, \frac{W}{2^s}$

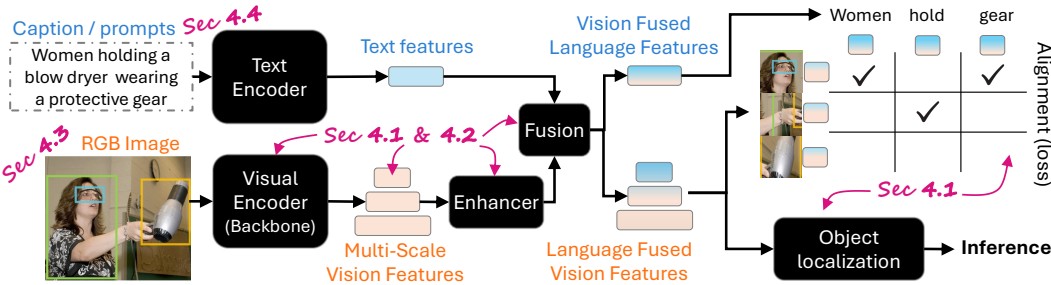

Figure 4: **Simplified Open-vocabulary Object Detector:** Trainable modules (black), may include additional modules and losses. Text features are fused with Multi-scale Vision features (last 4 blocks of visual encoder) via cross-self-attention, exchanging information across text and vision modality. The role of each component in robustness against noise is described in listed sections. The vision feature enhancer is commonly referred to as neck / FPN. Image modified from GLIP (2022).

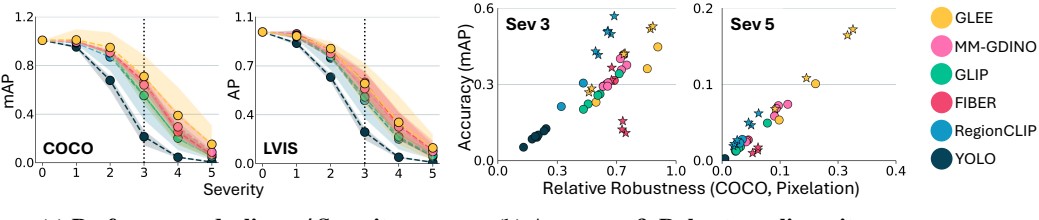

(a) **Performance decline w/ Severity**  (b) **Accuracy & Robustness linearity**

Figure 5: *(a)* Models start dropping performance around severity 3 ($\frac{H}{2^3}, \frac{W}{2^3}$). Shaded region encompasses the distribution of accuracy across all models, while solid line is mean accuracy. *(b)* Approx. linear relationship between accuracy and robustness preserves the ranking of models (robustness $\propto$ zero-shot accuracy). Fine-tuned models shown as stars. Both *(a)* & *(b)* are performed on pixelation.

the input image, leaving *textual captions unchanged*. Higher severity (4, 5) risks random predictions ($Acc_{Noise} \simeq 0$, **fig. 5a**). Thus, we judiciously use these only to support certain observations.

**Evaluation:** Metrics include AP (LVIS), mAP (COCO), and $AP_{avg}$ (ODinW-13). We shall use relative robustness (Chen et al., 2024; Schiappa et al., 2024) or 'robustness' as the key metric for measuring a model's robustness against noise. Relative Robustness = $1 - (\text{Drop in Accuracy/Accuracy}) = 1 - (Acc_{Clean} - Acc_{Noise})/Acc_{Clean}$. Here, $Acc_{Clean}$ and $Acc_{Noise}$ denote accuracy on original and noisy images. Relative Robustness is independent of absolute performance, enabling cross-model / cross-dataset comparisons. We also observe a linear relationship between absolute accuracy and relative robustness (**Figure 5b**). The outliers are mostly fine-tuned models, namely RegionCLIP (★, accuracy ↑ & robustness ↑), and FIBER-B (★ accuracy ↓ & constant robustness).

## 4 ANALYSIS

In the following sections, all the model variants shall be discussed, with special emphasis on zero-shot ones. All analyses Y-axis will almost always represent robustness. Every observation is highlighted, with **[Takeaways & Model Design]** describing the insight to design a robust OV-OD.

### 4.1 MODEL-BASED ANALYSIS

**Figure 6a** shows a strong positive correlation between robustness and model size (entire detector + text backbone) with pearson correlation on COCO / LVIS: 0.68 / 0.66 for pixelation, 0.78 / 0.72 for turbulence, and 0.77 / 0.70 for motion blur[2]. Transformer detectors consistently outperform CNNs, especially GLEE's Swin-L & EVA-02-L variants higher robustness vs ResNet variant ($< 19.4$ size).

**Figure 6b** groups detectors by vision backbone (irrespective of modules like enhancers, fusion, decoders, language backbones *etc.*), revealing: 1) Models with similar backbones show **consis-**

---

[2]Pearson correlation $< 0.3$ is none / weak and moderate for $[0.3, 0.7]$

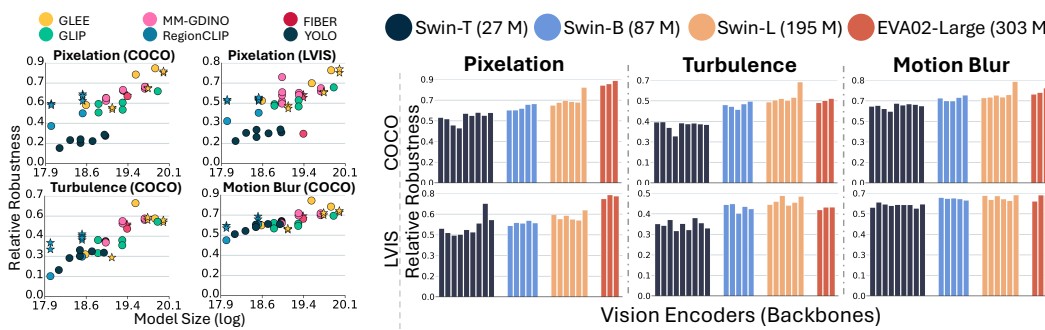

(a) **Size & Robustness** +ve correlation

(b) **Detectors grouped with similar vision backbones**

Figure 6: *(a)* Larger models are more robust (*e.g.* GLEE), while ResNet ones (RegionCLIP, YOLO, and GLEE variant) are least. Fine-tuned models as ★. *(b)* Performance remains relatively consistent across models with similar backbones. EVA-02 (303M) & Swin-L (195M), 24-blocks, have $\simeq$ robustness to Swin-B (12-blocks, 87M) on turbulence/motion blur. # of Parameters *'M'* is millions.

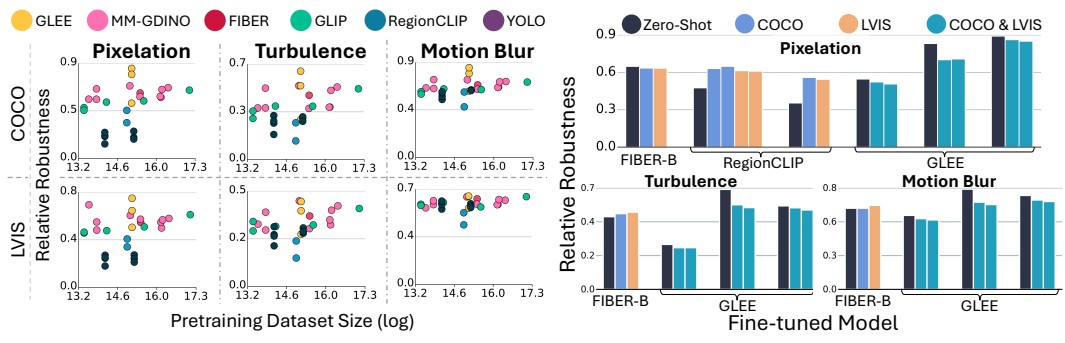

(a) **Pretraining dataset** of zero-shot models

(b) **Fine-tuning w/o distortion**: COCO eval.

Figure 7: *(a)* No clear correlation, GLIP (green) robustness consistent regardless of pretraining size. *(b)* Finetuning (color $\implies$ finetuning dataset) improves RegionCLIP & FIBER-B, but hurts GLEE.

**tent robustness** across different configurations. 2) Larger backbones are almost always robust (*e.g.* EVA-02). However, in general (turbulence & motion blur), depth of transformers is **not** crucial, *i.e.* ResNet < Swin-T (12 blocks, 27M) < Swin-B (12 blocks, 87M) $\simeq$ Swin-L (24 blocks 195M) $\simeq$ EVA-02 (24 blocks 303M), where $\simeq$ indicates similar robustness. Similar trend observed on other noises like Pixel drop, ISO, Salt-pepper, JPG compression, and Fog (*Supplementary*).

**Figure 7a** shows pre-training dataset size weakly affects the zero-shot robustness, which indirectly means different pre-training datasets (& number of pre-training datasets). Example, GLIP show steady robustness across different pretraining sizes (green dots). Pearson's correlation on COCO / LVIS: 0.34 / 0.28 for pixelation, 0.37 / 0.37 for turbulence, and 0.34 / 0.23 for motion blur.

**Figure 7b** evaluates the robustness of the COCO and LVIS fine-tuned models, with RegionCLIP and FIBER-B gaining robustness, and adversely affecting GLEE. When fine-tuning significantly improves performance, we observe a gain in robustness (RegionCLIP and FIBER-B); however, for GLEE, this gain is minimal, resulting in no meaningful impact on robustness. Overall, fine-tuning is **not** a universal robustness boost. Additionally, fine-tuning on COCO and LVIS (same images, different annotations) has similar improvement/degradation for robustness, *i.e.* impact of fine-tuning is influenced by the domain of images, while annotation plays a minimal role.

**[Takeaways & Model Design]** The robustness gap between ResNets (50 M) and large transformers like Swin-L or EVA-02-L (195–303 M) is driven mainly by backbone size. GLEE's ResNet variants remain less robust than transformer counterparts even with the same pipeline. The choice of backbones decides the robustness, while other bells and whistles, like additional modules (*e.g.* MM-GDINO/GLEE decoder), and extensive pre-training (*e.g.* MM-GDINO pre-trained on 9 datasets, and GLEE on 18 via three stages of training *etc.*) play a minimal role. Given resource constraint environments, Swin-B can be an amazing alternative to EVA-02 (4x bigger), and Swin-L (2x big-

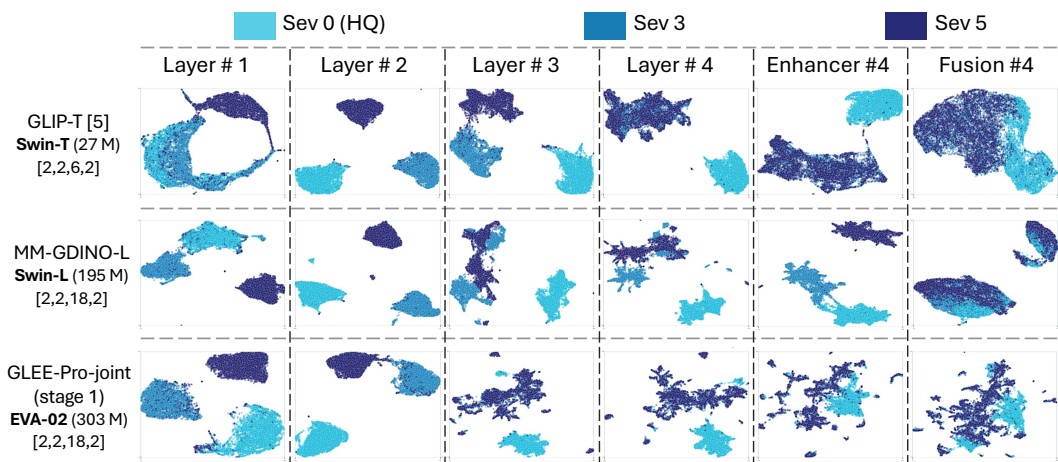

Figure 8: **Pixelation Features UMAP:** Multi-scale vision features, '#4' refers to the last layer (layer #4 at 24-th blocks for Swin-L/EVA-02, and 12-th for Swin-T), last layer enhanced features (Enhancer #4), and language fused features (Fusion #4). Dark patches (sev 5) overlapping with lighter ones (sev 0) indicate sev 5 $\simeq$ sev 0, *i.e.* robustness. Other noises in *supplementary*

ger), with *size/depth playing a limited role* for sufficiently large backbones. Model design should also incorporate explicit noise-robust training, without relying on pretraining and fine-tuning.

## 4.2 DEEPER INSIGHT INTO TRANSFORMERS: GLIP VS MM-GDINO VS GLEE

Figure 6b showed consistent robustness for Swin-L in GLIP, MM-GDINO, and GLEE, with a maximum difference in robustness of 0.15/0.08 in COCO/LVIS in all noise. This narrows down the analysis to the feature of the vision backbone. **Figure 8** illustrates the UMAP (2018) of the last 4 layers of the backbone, last layer feature for feature enhancer and fusion network. Ideally, the models shouldn't distinguish between severities, *i.e.* sev 5 features behave like (overlap) HQ sev 0 features. Explanation with t-SNE plots and other noises is provided in *Supplementary*.

**1) Backbone (Layer #1,#2,#3,#4):** Deeper layers overlap features across severities more (sev 3 & sev 5), showing that early layers are more vulnerable to noise. Shallow layers (# 1 & #2) at the **same depth (2 & 4 blocks) have similar feature collapse across all models**, despite architectural differences, partially explaining why similar depth backbones have identical robustness (fig. 6b).

**2) Enhancer:** For all models, the feature enhancer is just a convolutional block on backbone features. From a robustness perspective, feature enhancer serves no utility (part of "bells and whistles") *i.e.* 'enhancement' of backbone last layer does not affect overlap of severities.

**3) Fusion:** Fusion cross-exchanges vision features (across receptive fields / layers) with language. While language does not significantly impact robustness (sec 4.4), information exchange between spatial tokens for all layers $192 \times 192, 96 \times 96, 48 \times 48, 24 \times 24$, induces robustness in the last layer $(24 \times 24)$, as evidenced by the significant overlap between features of sev 5 and sev 0 for all models.

[Takeaways & Model Design] Cross-exchange of information between vision layers should help impart robustness across layers. However, validating this feature exchange design is beyond the scope of our analysis (computational infeasibility, section 5). Backbones at **similar depth have similar feature collapse**, partly explaining why similar backbones have similar robustness.

## 4.3 ROBUSTNESS AS A FUNCTION OF DATASET

**Figure 9a** illustrates larger ($\geq 96 \times 96$) object detection is more robust to noise than the smaller ones ($\leq 32 \times 32$). **Figure 9b** illustrates that almost all **detectors are highly robust when there is only one object to detect**. As the number of objects/image goes above 3, robustness starts to see a drop, eventually saturating around 10 or more objects/image. The jitter in robustness after 29 objects likely stems from the small sample size in that bin. **Figure 9c** illustrate the IOU of overlapping ob-

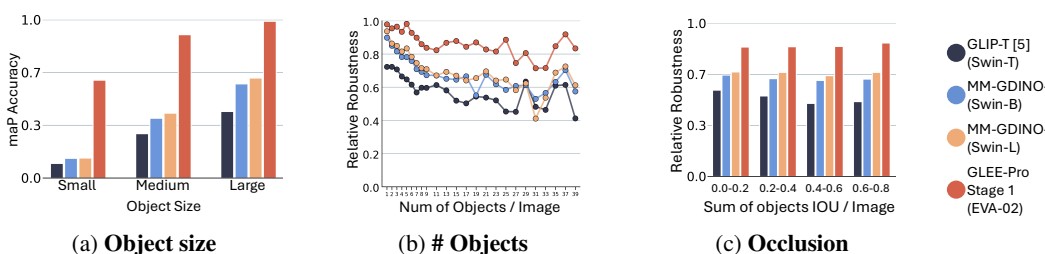

(a) **Object size**      (b) **# Objects**      (c) **Occlusion**

Figure 9: *(a)* Larger objects are more robust. *(b)* Models are very robust when the # of objects in an image is $< 3$, the jumps after $> 25$ objects are likely due to very few samples in that range. *(c)* Constant robustness across degrees of overlap between objects (not a function of occlusion). *(a,b,c)* are zero-shot models on COCO for pixelation. Other noises in *supplementary*.

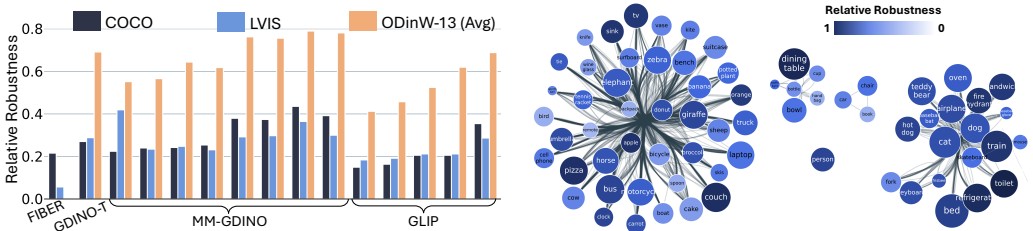

Figure 10: *(left)* **Dataset Dependent Robustness** Average ODinW-13 is more immune to noise than COCO & LVIS (comparable robustness) for transformers on pixelation (sev 4). *(right)* **Class-wise robustness** Some COCO classes are more robust (shade of blue) than others, with moderate correlation with object size (dot size). Classes grouped in bins of frequency (log) for GLIP-T.

jects *i.e.* occlusion, has hardly any impact on robustness (*counter-intuitive*). Cluttering (or business) in an image is a function of both the number and the occlusion of objects. Hence, it's safe to say that cluttered images are indifferently affected. Models follow the rankings in fig. 6b (pixelation).

**Figure 10** (left) shows ODinW-13 maintains high robustness scores of $\simeq 0.6$ across models, nearly twice that of LVIS and COCO, which exhibit similar robustness. This supports fig. 7b, confirming robustness relies more on the image domain (same images for COCO & LVIS) rather than the annotation type. Key differences: **1) Object Size:** Only $\approx 10\%$ of ODinW-13 objects are very small ($\leq 32 \times 32$) versus $\simeq 42\%$ in COCO and $\simeq 58\%$ in LVIS (fig. 9a). **2) Objects Density** ODinW-13 has $\simeq 50\%$ images with single object, versus $\simeq 12\%$ for COCO and $\simeq 38\%$ for LVIS (fig. 9b).

**Figure 10** (right) shows that robustness varies by object class, moderately reflected by average object size (size of dot). For pixelation, robustness correlates with mean class size for COCO / ODinW-13 as 0.52 / 0.45. On COCO, categories like *parking meter*, *stop sign*, and *toilet* are easiest to detect, while for ODinW-13 classes like *lobster*, *jellyfish*, and *hand* are easiest *(Supplementary)*. In contrast, robustness shows almost no correlation (Pearson) with class frequency (r$\simeq 0.02$ for COCO, $\simeq 0.021$ for ODinW-13), suggesting how often a class appears does not drive robustness. This explains why LVIS's long-tail distribution, simply adds rare classes to COCO, exhibits similar robustness to COCO. A possible explanation for the disproportionate robustness of certain objects is that they usually appear alone (*e.g.* a single traffic signal), and may be easier to detect (fig. 9b).

**[Takeaways & Model Design]** Models are inherently robust w/o a critical need for explicit robustness for large, singular objects. Conversely, datasets like ODinW-13 can thus overstate robustness, rather reflecting models' true robustness. Robustness seems to largely depend on diverse image domains rather than annotation (detecting *on* what is more important than what, COCO $\simeq$ LVIS).

## 4.4 EXPRESSIVENESS OF CAPTIONS AND PROMPT ENGINEERING

**Figure 11a** shows that fine-tuning on the REC datasets (RefCOCO, RefCOCO+, and RefCOCOg) results in similar robustness with a minor drop in performance for RefCOCO+. Empirically, this implies the descriptive nature (expressiveness) of captions or text prompts used in training has seem-

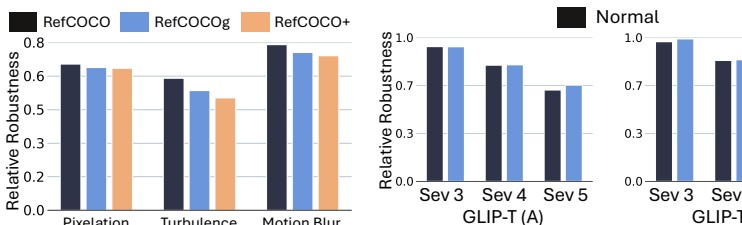

(a) **Robustness vs 'Train' Captions**   (b) **Prompt Engineering: Test Captions with pixelation context**

Figure 11: *(a)* REC fine-tuned FIBER evaluated on COCO. Despite *training* on captions with different degrees of expressiveness (RefCOCOg is most descriptive), robustness varies slightly, indicating limited impact on robustness. *(b)* *Evaluation* on Flickr30k with test captions modified with textual context of pixelation (light) (vs original (dark)), has minimal impact on robustness of GLIP variants.

ingly weak/limited impact on robustness. REC datasets are based on COCO images with differently phrased captions; RefCOCO contains simple expressions, while RefCOCO+ has appearance-based prompts, and RefCOCOg includes more elaborate and detailed language.

**Figure 11b** evalautes GLIP variants on Flickr30k via 2 sets of captions: 1) Normal: Original captions from the dataset (dark). 2) LLM-generated: Original captions fed into an LLM (Mishra et al., 2024) generating five augmented variants, infused with context of pixelation/low-resolution (light). For example, "*A boy smiles in front of a stony wall in a city*" becomes "*In a low-resolution cityscape, a boy's smile is captured against a rough stone wall*". Evaluation occurs on the average of vision embedding fused with 5 LR captions. Modified caption variants exhibit consistent robustness across severity levels across models; thus, it's safe to say prompt engineering test captions to introduce noise-awareness have almost no impact.

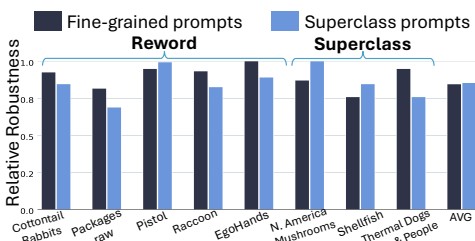

Figure 12: **Similar robustness for fine-grained vs Superclass:** GLIP-T on ODinW-13 with pixelation. Datasets with accuracy $\simeq 0$ not shown, 1-class datasets 'reworded' & multiple classes clubbed as 'superclass'.

**Figure 12** compares ODinW-13 *evaluation* using descriptive fine-grained prompts with coarse-grained superclass prompts (multiple classes grouped), and reworded class labels (dataset with 1 class). The small difference in robustness (mean $\Delta$ of $\simeq 0.14$ for pixelation, $\simeq 0.17$ for turbulence, $\simeq 0.12$ for motion blur, and negligible $\Delta$ overall) can be attributed to lower performance on coarse-grained superclass annotations (linearity between robustness & accuracy (fig. 5b)). On average, superclass annotations have limited impact on robustness. We only analyze 8 / 13 ODinW-13 datasets, ignoring cases where accuracy $\simeq 0$ (random).

**[Takeaways & Model Design]** The above findings suggest that expressive captions (during fine-tuning), and prompts engineered with the context of degradations (during evaluation) do not significantly improve robustness. Combining these, with previously observed results like the marginal impact of text backbone (fig. 6b), and annotations (section 4.3), its safe to say that **expressiveness or distortion-aware augmentation of prompts has minimal influence on 'visual' robustness**. To our knowledge, we are among the first to demonstrate this limited role of language, hinting that future robustness efforts targeting vision modality is likely to yeild more success. Another potential direction is to *re-train* models with prompts injected with noise context, rather than only at evaluation. This would require a new caption–image alignment and is therefore *left as future work*.

## 5   VALIDATION OF OUR ANALYSIS: LR-TK0+ & LR-TK0++

Our analysis has highlighted **two** key model designs: **1)** Cross-exchanging information across backbone layers: This requires architectural redesign, which requires re-training and is computationally very expensive *e.g.* GLEE was trained on 64 GPUs across 18 datasets. We leave this as *future work*. **2)** Visual backbone is the primary determinant of robustness, with shallow layers getting most adversely affected. We shall ignore text modality and other components (neck, fusion, *etc.*) to motivate a *lightweight* robustness solution for *shallow layers*, catered to real-world deployment (*zero-shot*).

Table 1: Ablation of Continual Learning.

| Distortion on LR-TK0+ | COCO mAP | COCO RR | ODinW-13 avg AP | ODinW-13 RR |
|---|---|---|---|---|
| Uniform | 27.81 | 0.60 | 37.78 | 0.90 |
| Background | 27.72 | 0.59 | 36.84 | 0.88 |
| Random | 27.35 | 0.59 | 36.64 | 0.87 |
| GT boxes (LR-TK0++) | **28.37** | **0.61** | **38.21** | **0.91** |

Table 2: GLIP-T Results on COCO (mAP) & Real-World Wider face (Yang et al., 2016). LR-TK0+ is modified LR-TK0 for hierarchical transformers

| Model GLIP-T [5] | Backbone # Params | COCO (Pixelation) Sev 1 | Sev 2 | Sev 3 | Wider face AP | AP50 |
|---|---|---|---|---|---|---|
| Zero-shot | 27.5 M | **46.0** | **42.0** | 26.2 | 12.57 | 26.00 |
| LR-TK0+ | 29.1 M | 44.5 | 41.4 | 28.2 | 13.32 | **29.88** |
| LR-TK0++ | 29.1 M | 45.5 | 41.7 | **28.4** | **13.99** | 29.88 |

**Background:** Improving robustness ideally shouldn't rely on the test domain, nor should it lose the generalization of VLMs. Previous approaches include RobustSAM (Chen et al., 2024), Super-Resolution (Gao et al., 2023), Test-Time Adaptation (Hakim et al., 2025) *etc*. However, these approaches are either *computationally too heavy* for **OV-ODs** ($\approx$3-5 transformer) or require the *knowledge of the test domain*. Distillation needs two OV-ODs, Super-Resolution is already too heavy in addition to OV-OD (& unreliable zero-shot super-resolution), and Test-Time Adaptation trains on test data (and modifies pretrained weights). Efforts rarely overlap with object detectors (because of complexity), and are restricted to smaller CNN-based Faster-RCNNs or less complex tasks.

**LR-TK0+:** We extend LR-TK0 (Pathak et al., 2025), an existing approach for low-resolution classification (preserves zero-shot), to hierarchical transformers (*e.g.* Swin) for object detection. Unlike the original LR-TK0, which uses a costly distillation setup, we only retain the cost-efficient trainable tokens and remove the teacher-student design. At *every* layer (especially shallow layers), we insert a fixed $32 \times 32$ set of *trainable spatial tokens*, interpolated to match the layer's spatial resolution and added to the *frozen* feature maps. Interpolating a small number of tokens has two advantages: 1) *Flexible hierarchy*: Original LR-TK0 attaches a fixed number of prompts to every ViT token, thus cannot accommodate varying $H \times W$, while interpolation can match varying spatial resolution per layer. 2) *Lower overhead:* For a $600 \times 600$ Swin-T input, our $32 \times 32$ tokens add only 5.7% parameters, compared to 22.5% for fixed-token design. We denote this extension as **LR-TK0+**.

**LR-TK0++:** On top of LR-TK0+, we introduce a lightweight & cost-efficient *continual learning* strategy to mimic real-world degradations. Unlike distillation (which requires two passes for teacher & student), continual learning has no additional cost (single forward pass). Training begins with clean images and gradually introduces pixelation *inside ground-truth boxes* for the first $T_1 = 10$ epochs, with linearly increasing probability. This helps the model detect pixelated foreground objects against familiar clean HQ backgrounds. After $T_1$, we progressively perturb regions *outside* GT boxes, again with increasing probability, while continuing to sample pixelated GT boxes. This targeted curriculum, denoted **LR-TK0++**, encourages robust object features under degradation, outperforming uniform or random, or background perturbation (table 1).

**Results.** Trained on a 3,000-image Flickr30k subset, LR-TK0+ and LR-TK0++ improve robustness on COCO and the real-world Wider Face dataset under pixelation (table 2). Trained for severe 3 pixelation, LR-TK0+ shows slight drops at lower severity of noise (1 & 2), but the LR-TK0++ approach of gradually transitions from HQ-to-pixelated images, minimizing this drop. LR-TK0++ shows gains over the random augmentation approach of LR-TK0+ while maintaining efficiency. It's to be noted that the entire GLIP-T was frozen for this training, only impacting the visual backbone. These results illustrate how insights from our analysis can inspire cost-efficient extensions to improve robustness in real-world noisy conditions. More in *Supplementary*.

## 6 CONCLUSION

**Robust Onion** provides a detailed analysis of robustness against visual distortions in OV-ODs, systematically *'peeling'* each component, to assess their individual impact. Analyzing state-of-the-art detectors under realistic, yet underexplored, visual distortions (mirroring real-world noise feature collapse) reveals: 1) Vision backbone dominates robustness, outweighing architectural variations or pre-training choices. 2) Shallow layers are most noise-sensitive, and datasets biased toward large, isolated objects can falsely suggest high robustness. 3) Prompt or caption expressiveness has minimal effect on robustness during both inference and fine-tuning. Our results include promising directions such as continual learning, LR-TKO++, and cross-layer information sharing. Robust Onion provides a clear, actionable roadmap for advancing the robustness of next-generation OV-OD.

## 7 REPRODUCIBILITY STATEMENT

All the models used in this work are publicly available (open-source githubs), with results (HQ results) verified from their official GitHub and paper. The code for reproducing all the noises mentioned in the paper is also open source and will additionally be made available with this paper GitHub upon acceptance, along with all results. All analysis done for pixelation can simply be replicated by 1 line of code for resizing : $(H, W) \rightarrow (\frac{H}{8}, \frac{W}{8}) \rightarrow H, W$. Models were evaluated on 1 48GB GPU on A6000 (Ampere) GPUs. Our proposed research direction code is taken from LR-TK0 (Pathak et al., 2025) open source code, with GLIP (Li et al., 2022) open source code used as a base to train GLIP-T model. No hyperparameters were modified, and the code simply adds the tokens to the Swin-T backbone.

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

## A APPENDIX

## A MOTIVATION

Image perturbations significantly affects the performance of the detection models. As, we increase the severity, models often misclassify objects (Figure 13) and fail to preserve accurate bounding box predictions (Figure 16). Some samples of the perturbations are shown in Figure 18. Some sample detections on images without synthetic perturbations is shown in Figure 17. This shows the model fails to detect accurately even on the most prominent class (person).

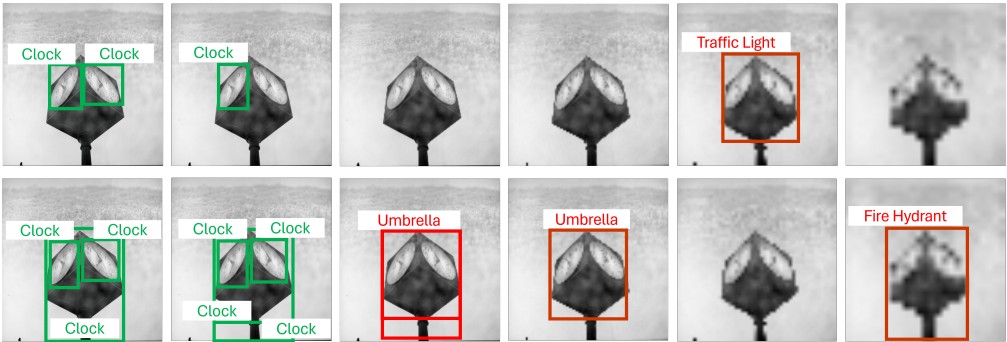

Figure 13: **Progressive Pixelation:** GLIP (Li et al., 2022) (top) and MM-GDINO (Zhao et al., 2024b) (bottom); performance degrades on COCO image (888 × 924) from left (clean) to right (pixelated) via downsampling by $\frac{1}{2}$.

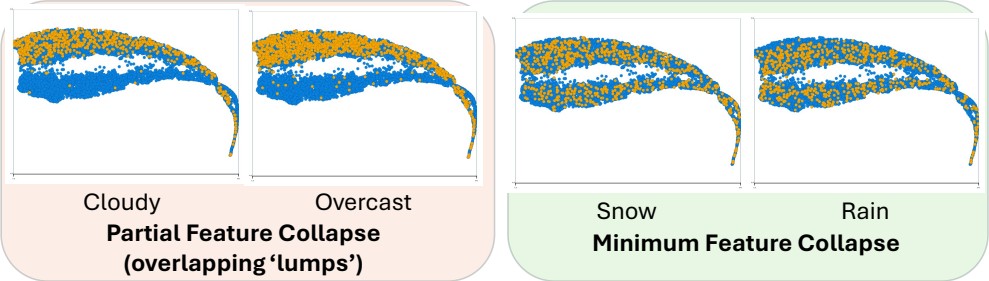

Figure 14: **Real World** BDD-100K all Feature Collapses.

Figure 16 is the continuation of Figure 13 indicating how as severity of pixelation is increased, GLIP detection ability degrades. Two visible degradations are: 1) Model can't detect multiple objects instead clubs them all under one detection at lower severity 2) Model loses the capability of detecting small sized objects.

## B MODEL ZOO

In Table 3, we present the details of all the benchmark models considered, including their visual backbones, sizes, and the corresponding pre-trained datasets along with their sizes. Region-CLIP (Zhong et al., 2022) and GLEE (Wu et al., 2024) are the models which has a ResNet-based visual backbone. On the other hand, models like FIBER (Dou et al., 2022), GLIP (Li et al., 2022), MM-GDINO (Zhao et al., 2024b), and GLEE (Wu et al., 2024) leverage Swin Transformers. Notably, the GLEE (Wu et al., 2024) model uses `EVA-02 Large` backbone, which is the largest backbone considered in the study and greatly contributes to the higher robustness.

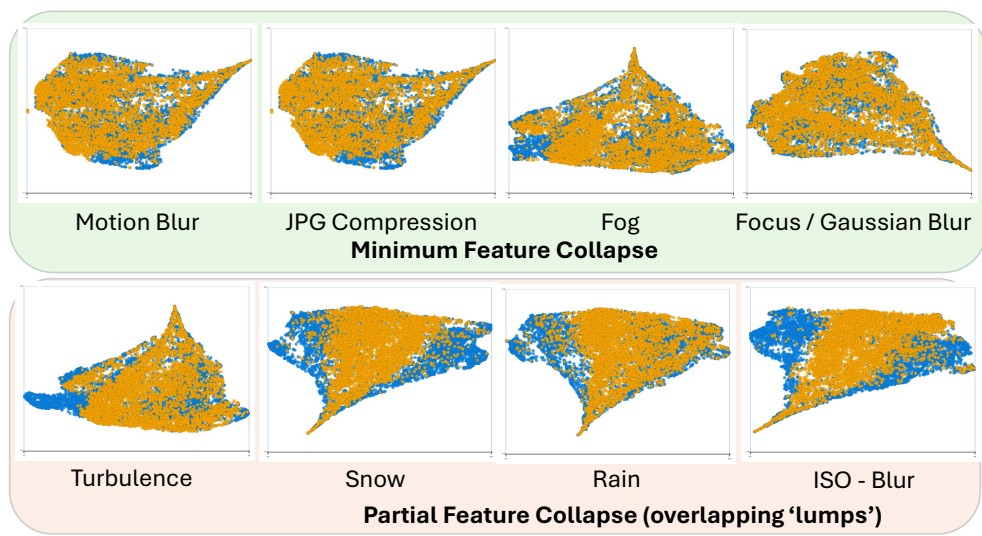

Figure 15: **Synthetic Noises mimicking Real World** all Feature Collapses.

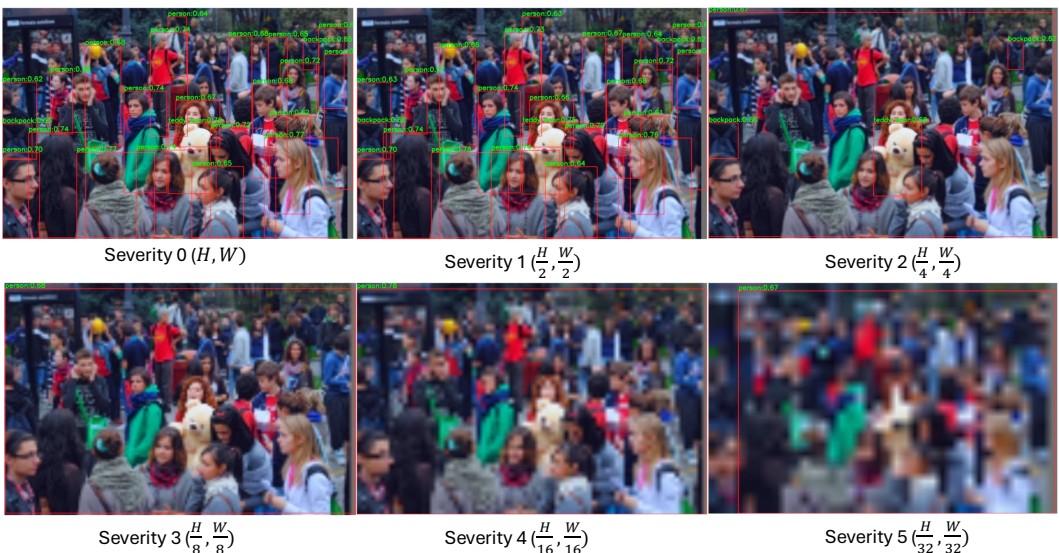

Figure 16: **Models progressive degradation with pixelation.**

## C    DATASET DESCRIPTIONS

We evaluate the zero-shot performance of object detectors on three standard benchmarks to analyze robustness against pixelation: **COCO** (Lin et al., 2015) (val2017): Contains 5,000 images with 80 object categories. The validation set includes approximately 36,781 object instances. **LVIS** (Gupta et al., 2019; Kamath et al., 2021) (MiniVal): A long-tail detection dataset comprising 1,203 object categories. The MiniVal set contains 5,000 images with about 62,397 object instances. **ODinW-13** (Li et al., 2022): A collection of 13 small out-of-distribution datasets, totaling approximately 3,235 images across diverse domains.

It's important to note that COCO and LVIS share the same image set but differ in their annotations and train/val/test splits. Regarding object categories, LVIS can be considered a superset of COCO, with COCO's 80 categories being a subset of LVIS's 1,203 categories. This relationship allows for interesting cross-dataset comparisons and analyses.

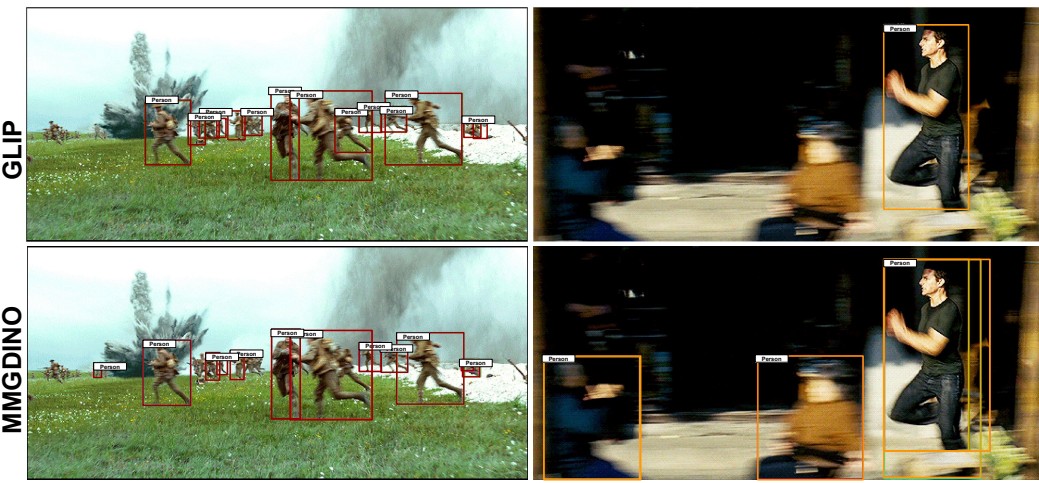

Figure 17: **Sample detections results on real world images collected from internet without synthetic perturbations.**

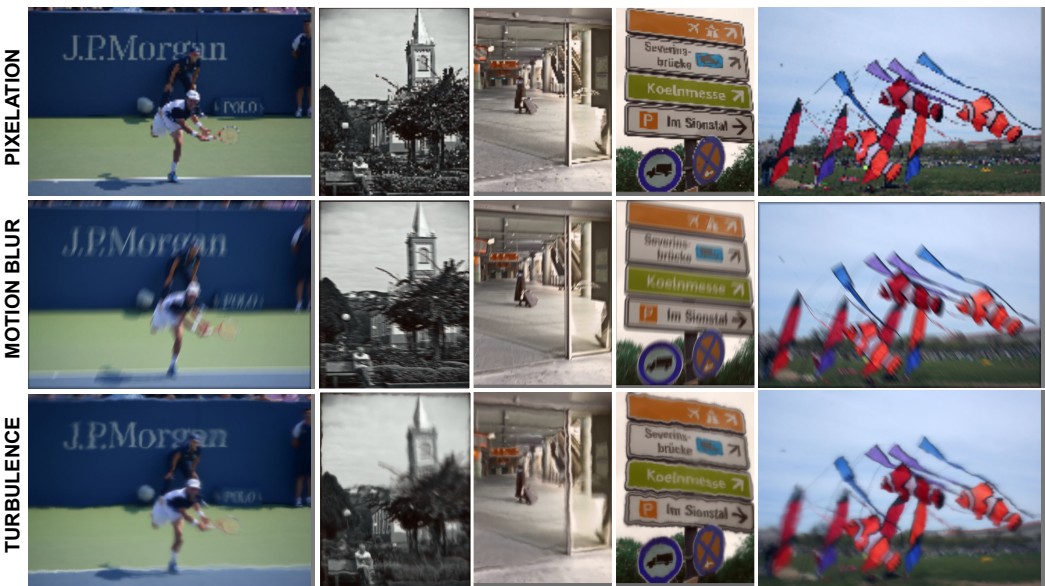

Figure 18: **Samples from noise perturbations**

For deeper insights and training our proposed solution, we utilize the **Flickr30k Entities** (Plummer et al., 2015) dataset, which contains 31,783 images and 275,775 bounding boxes. This dataset is commonly employed in pretraining zero-shot models (referred to as "Gold") or fine-tuning them (referred to as "MDETR" data).

**Wider face** (Yang et al., 2016) has a lot of tiny faces (pixelation when resized to 224 x 224) in a variety of real-world settings. We used the validation set of Wider face dataset for evaluation, which contains 3226 images and 39496 annotated faces.

C.1    REFERRING EXPRESSION COMPREHENSION TASK (REFCOCO, REFCOCO+, REFCOCOG)

Referring Expression Comprehension (REC) is a task, in which, given an image and an expression (for example, *"A red colored ferrari"*), the model should detect the region corresponding to the

Table 3: **Benchmark Models (6 Models and 45 Backbones):** Pre-training is image-text pairs from datasets like Object365 (Shao et al., 2019), OpenImages (Krasin et al., 2017), GoldG (Kamath et al., 2021), CC (Sharma et al., 2018), *etc*. Visual Backbone uses Swin-Transformer (Liu et al., 2021) (mostly) and ResNets (He et al., 2015).

| Models | # Backbones and Size (in Million) | | Pretraining Datasets and Size (in Million) | |
|---|---|---|---|---|
| **RegionCLIP** | 8 ResNets (RN50 & RN50x4) | 65-114 | CC3M, COCO Caption | 3-3.1 |
| **FIBER** | 6 Swin-Transformers (Swin-Base) | 252 | COCO, CC, SBU, VG | 13 |
| **GLIP** | 5 Swin-Transformers (4 Swin-Tiny & 1 Swin-Large) | 152-430 | O365, GoldG (Flickr30K+VG+GQA) CC3M, SBU, CC12M, OI | 0.66-27 |
| **MM-GDINO** | 9 Swin Trans (5 Swin-Tiny, 2 Swin-Base, 2 Swin-Large) | 174-343 | O365, GoldG, OI, GRIT, V3Det,COCO, RefCOCO, RefCOCO+, RefCOCOg | 1.7-13 |
| **YOLO** | 7 YOLOv8(1 YOLOv8-S, 1 YOLOv8-M, 3 YOLOv8-L, 1 YOLOv8-X, 1 YOLOv8-XL) | 76-168 | O365, GoldG, CC3M | 1.4-1.6 |
| **GLEE** | 3 ResNets (RN50), 3 EVA-02 Large, 3 Swin Transformers (Swin-Large) | 121-476 | Stage-1: O365, OI; Stage-2: COCO, LVIS, BDD, YTVIS19, YTVIS21, OVIS, RefCOCO, RefCOCO+, RefCOCOg, VG , VOS , RVOS , UVO , UVO-dense ; Stage-3: SA1B , GRIT | Stage-1: 3.6 Stage-2: 0.9 Stage-3: 7.3 |

expression. This task was evaluated on RefCOCO, RefCOCO+, and RefCOCOg, which is derived by the detail in the expressions.

RefCOCO: RefCOCO was collected using an interactive two-player game called ReferItGame, where one player described a target object in an image, and the other had to identify it. As a result, the referring expressions in RefCOCO are typically short and direct, averaging around 3–4 words. These expressions commonly include both appearance and spatial cues. Example: "The red and white checkered table on the left"

RefCOCO+: RefCOCO+ was also created using the same ReferItGame framework, but with one key restriction: annotators were not allowed to use absolute spatial terms (such as "left," "right," "top," etc.). This restriction forces the referring expressions to rely solely on appearance, attributes, and relative object descriptions, rather than location-based cues. Example: "The giraffe with lowered head".

RefCOCOg: Unlike RefCOCO and RefCOCO+, it was collected offline (not through a game), by making annotators write longer and more natural, descriptive, and contextual expressions. On average, expressions in RefCOCOg are around 8 to 9 words long, often including complex language, object relationships, and scene-level reasoning. Example: "An adult giraffe scratching its back with its horn".

**NOTE:** REC dataset results are not reported in the main paper, since they have abnormally high robustness scores. The reason behind the high robustness of the REC fine-tuned models is the low accuracy of FIBER-B REC finetuned models on COCO (and LVIS), which results in a small drop in accuracy on noises (random predictions remain random), giving "abnormally high robustness scores" (Pathak et al., 2025).

# D  ADDITIONAL FIGURE / DETAILS IN MAIN SUBMISSION

## D.1  FIGURE 3

Here we show the robustness scores for all models perturbed with atmospheric turbulence in Figure 19a and with motion blur in Figure 19b.

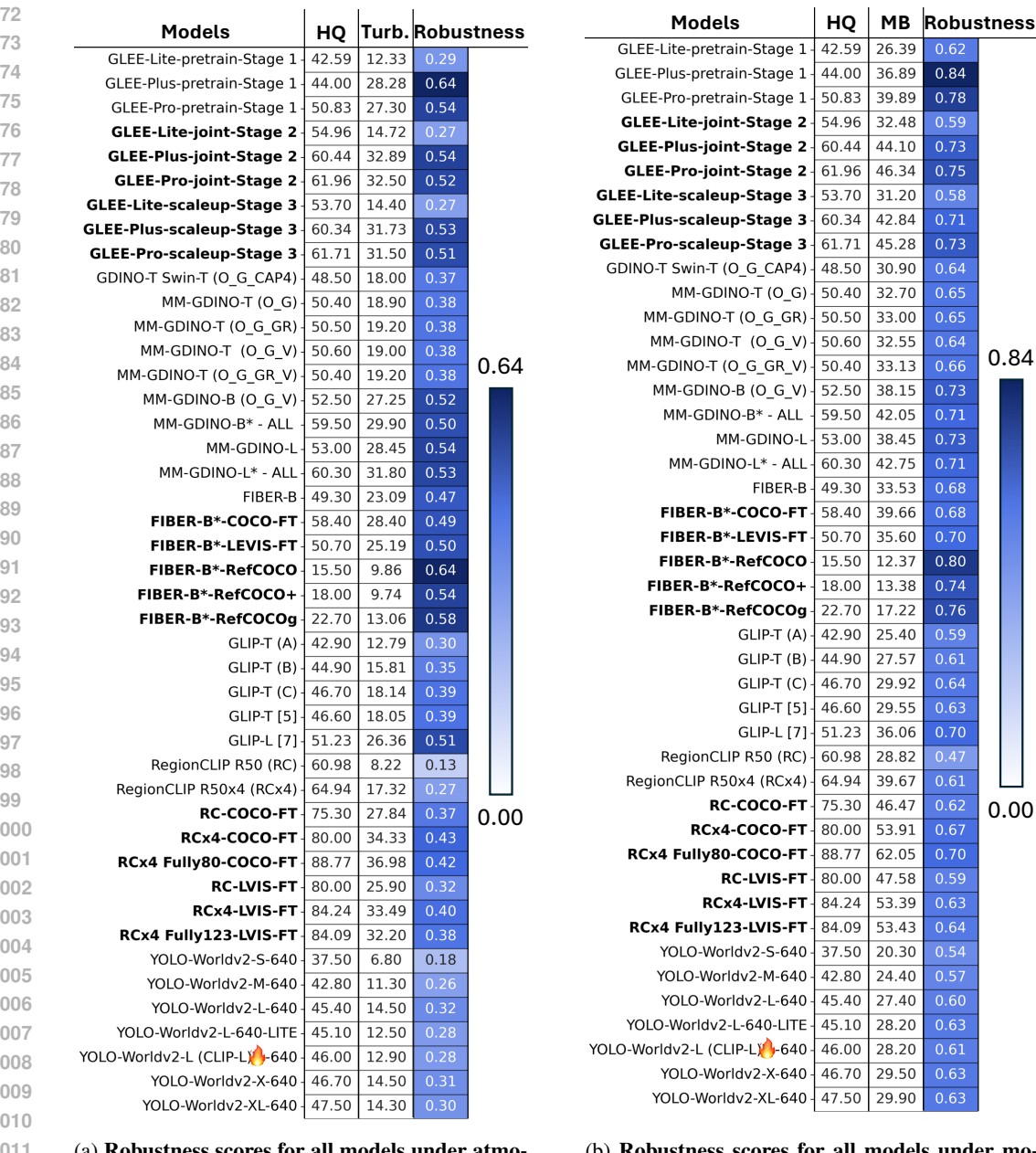

(a) **Robustness scores for all models under atmospheric turbulence perturbation**

(b) **Robustness scores for all models under motion blur perturbation**

Figure 19

## D.2 FIGURE 5B

Figure 5b showed results for COCO; here we show results for Accuracy vs Robustness for LVIS. A similar linear relationship between robustness and accuracy of **Zero-shot** detectors exists, except for fine-tuned models (shown in stars).

## D.3 FIGURE 12

Figure 12 shows only 8 datasets out of 13 OdinW-13 datasets. This is because either Fine-grained or superclass evaluation accuracy at sev 3 is so close to random prediction that it can't be reliably used to make any inference. Near random prediction models achieve abnormal robustness scores Pathak

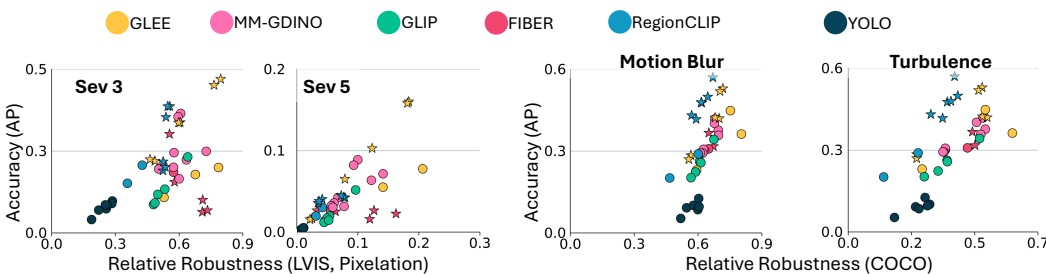

Figure 20: **Accuracy, Robustness linear relationship**. Same details as those of Figure 5b.

et al. (2025). The super class annotation was obtained from the official annotation of OdinW-13. The superclass for OdinW-13 datasets are as follows (super category written in brackets):

AerialMaritimeDrone (movable-objects), Aquarium (creatures), CottontailRabbits (Cottontail-Rabbit), EgoHands (hands), NorthAmericaMushrooms (mushroom), Packages (packages), PascalVOC (VOC), pistols (Guns), pothole (potholes), Raccoon (raccoons), Shellfish (shellfish), thermalDogsAndPeople (dog-person), VehiclesOpenImages (vehicles)

Some datasets don't have meaningful supercategories; hence, they were removed during evaluation. The datasets like AerialMaritimeDrone, PascalVOC, and Aquarium have supercategories that do not align with their class labels. Others, like Egohands, Packages, Raccoons, Pistols, and Cottontail Rabbits, have matching/similar supercategories and class labels because they have have only one class.

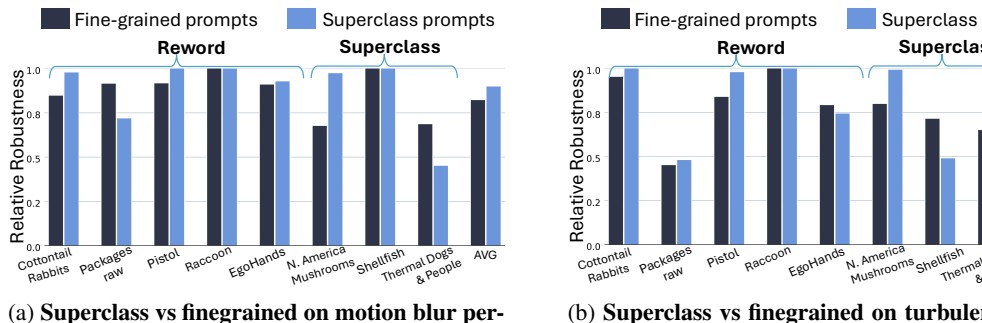

(a) **Superclass vs finegrained on motion blur perturbation**

(b) **Superclass vs finegrained on turbulence perturbation**

Figure 21: *(a)* shows the superclass prompting performance with the motion blur perturbation *(b)* shows the superclass prompting performance with the turbulence perturbation. This follows the same trend as the pixelation perturbation, where the superclass/finegrained prompting doesn't vary the performance.

### D.4 FIGURE 6A

Figure 6a showed results for COCO, here we show results for Robustness vs model size for LVIS at sev3 and sev5 in Figure 22. We also show the results for real world noises in Figure 23

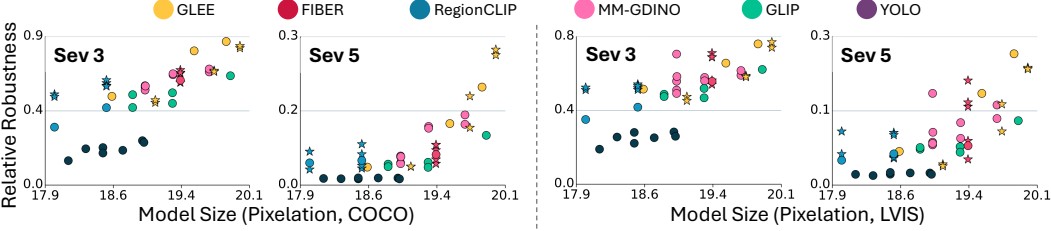

Figure 22: **Robustness vs model size for pixelation severities**. Same details as that of Figure 6a.

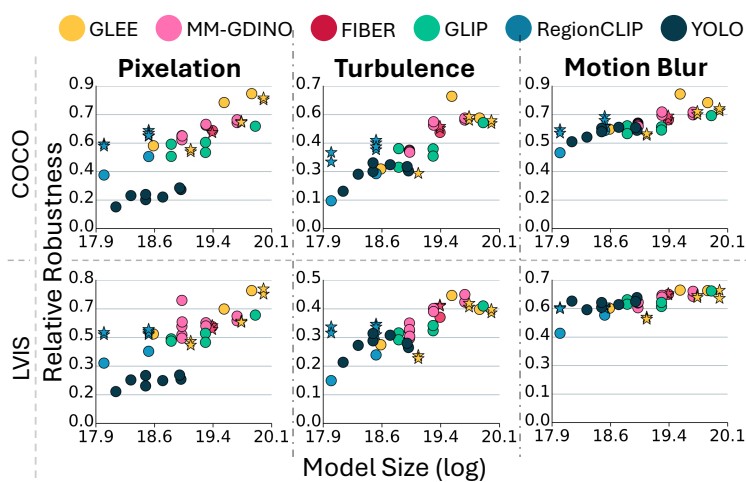

Figure 23: **Robustness vs model size for real world noises**. Same details as that of Figure 6a.

## D.5 FIGURE 6B

Figure 6b showed results for COCO & LVIS for sev 3, here we show results for sev 5. Performance is consistent across backbones here as well. Since accuracy is so low, close to random predictions, the outlier behavior can be not be used to draw reliable conclusions.

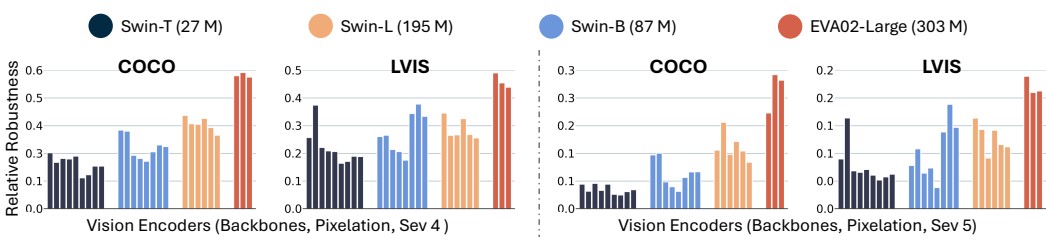

Figure 24: **Robustness vs Backbone Size at sev 4 and sev 5**. Same details as that of Figure 6b.

## D.6 FIGURE 7A

Figure 7a showed the results for effect of pretraining dataset size in robustness for all noises. Here we show the same trend for all severity in Figure 25

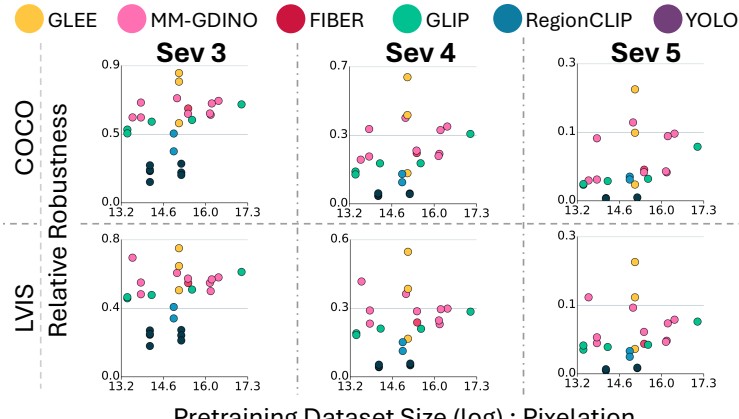

Figure 25: **Robustness vs Dataset Size for all severity**. Same details as that of Figure 7a.

### D.7 FIGURE 7B

Figure 7b showed the effect of finetuning on COCO and LVIS on robustness. Here we show the effect of finetuning for all noises in Figure 26 and for all noises in Figure 27

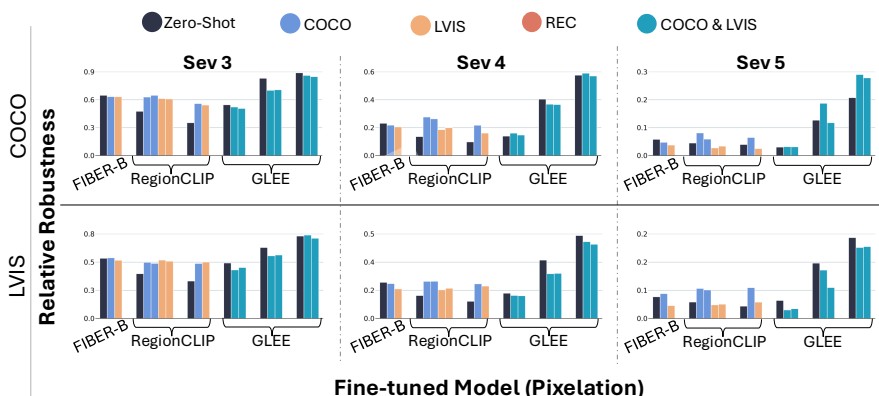

Figure 26: **Effect of finetuning on robustness across severity**. Same details as that of Figure 7b.

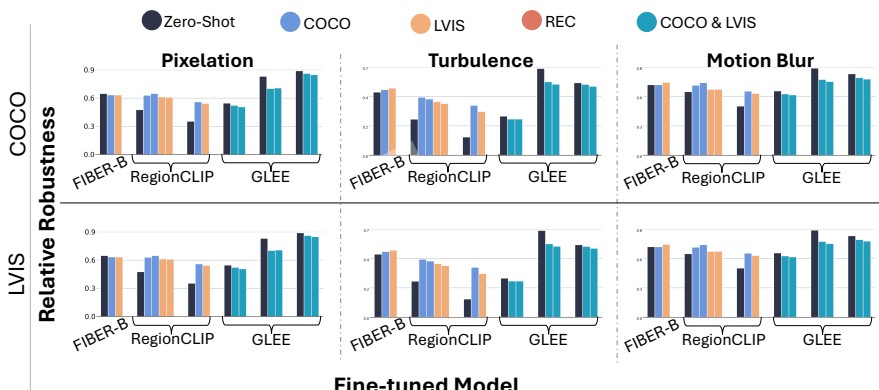

Figure 27: **Effect of finetuning on robustness across noises**. Same details as that of Figure 7b.

### D.8 FIGURE 8

Figure 8 showed, UMAP plot of features, here we show t-SNE plot for the same.

For **GLIP**, backbone features '$\mathcal{B}$' (the last 4 layers of backbone, $\mathcal{B}_1$, $\mathcal{B}_2$, $\mathcal{B}_3$, and $\mathcal{B}_4$) are used for Swin-T transformers with blocks partitions as [2,2,6,2]. These multi-scale features (features from different intermediate backbone layers) are passed through a neck network, which are simple intermediate convolutional layers (channels $\rightarrow$ channels), such as $\mathcal{C}(D \rightarrow 256)$ & $\mathcal{H}(256 \rightarrow 256)$, on these backbone features creating '$\mathcal{N}$' features as $\mathcal{N}_1, \mathcal{N}_2, \mathcal{N}_3, \mathcal{N}_4 \& \mathcal{N}_5$, where $\mathcal{N}_3 = \mathcal{H}_1(\mathcal{C}_1(\mathcal{B}_4)) \parallel \mathcal{N}_2 = \mathcal{H}_2(\mathcal{C}_2(\mathcal{B}_3) + \mathcal{C}_1(\mathcal{B}_4)) \parallel \mathcal{N}_1 = \mathcal{H}_3(\mathcal{C}_3(\mathcal{B}_2) + \mathcal{C}_2(\mathcal{B}_3) + \mathcal{C}_1(\mathcal{B}_4)) \parallel \mathcal{N}_4 = \mathcal{H}_4(\mathcal{C}_1(\mathcal{B}_4)) \parallel \mathcal{N}_5 = \mathcal{H}_5(\mathcal{H}_4(\mathcal{C}_1(\mathcal{B}_4)))$. The Fusion network induces text context into vision neck features, generating '$\mathcal{F}$' features as $\mathcal{F}_1, \mathcal{F}_2, \mathcal{F}_3, \mathcal{F}_4, \mathcal{F}_5$ for $\mathcal{N}_1, \mathcal{N}_2, \mathcal{N}_3, \mathcal{N}_4, \mathcal{N}_5$ respectively. For plotting, we use $\boldsymbol{\mathcal{B}_1, \mathcal{B}_2, \mathcal{B}_3, \mathcal{B}_4, \mathcal{N}_4}, \& \boldsymbol{\mathcal{F}_4}$.

For **MMGDINO** (Swin-Large), backbone features '$\mathcal{B}$' ($\mathcal{B}_1$, $\mathcal{B}_2$, $\mathcal{B}_3$, and $\mathcal{B}_4$) are used with 24 blocks partitioned as [2,2,18,2]. These multi-scale features are passed through a neck network, which produces neck '$\mathcal{N}$' features as $\mathcal{N}_1, \mathcal{N}_2, \mathcal{N}_3, \mathcal{N}_4$, and $\mathcal{N}_5$. Here, the neck is a simple convolutional network, $\mathcal{N}_1 = \mathcal{C}_1(\mathcal{B}_1) \parallel \mathcal{N}_2 = \mathcal{C}_2(\mathcal{B}_2) \parallel \mathcal{N}_3 = \mathcal{C}_3(\mathcal{B}_3) \parallel \mathcal{N}_4 = \mathcal{C}_4(\mathcal{B}_4) \parallel \mathcal{N}_5 = \mathcal{C}_5(\mathcal{B}_4)$. This extra neck feature $\mathcal{N}_5$ is termed as extra_convs in the original code. The Fusion network consists of an encoder-decoder structure, with early fusion, meaning the encoder fuses the textual feature in the

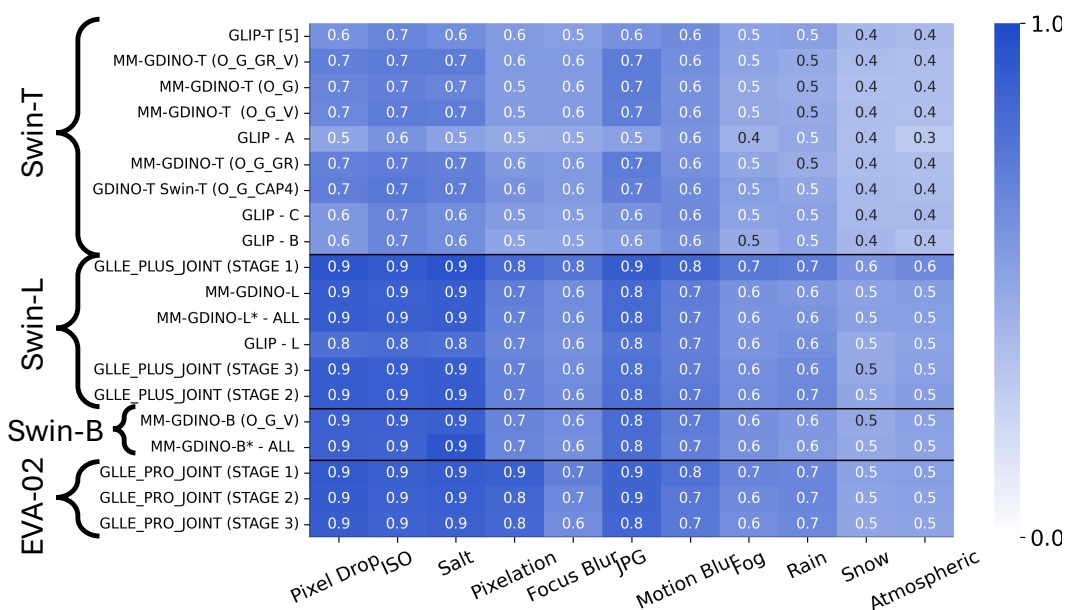

Figure 28: **All noises for all backbones.**. Same details as those of Figure 6b.

vision feature at both the encoder and decoder stages. To maintain uniformity with GLIP, we plot only encoder fused features as its much closer to the parameter size of GLIP fusion transformers. Fused features '$\mathcal{F}$' correspond to neck features as $\mathcal{N}_1 \to \mathcal{F}_1, \mathcal{N}_2 \to \mathcal{F}_2, \mathcal{N}_3 \to \mathcal{F}_3, \mathcal{N}_4 \to \mathcal{F}_4$, and $\mathcal{N}_5 \to \mathcal{F}_5$. For plotting, we use $\mathcal{B}_1, \mathcal{B}_2, \mathcal{B}_3, \mathcal{B}_4, \mathcal{N}_5, \& \mathcal{F}_5$.

For **GLEE**, model EVA-02 24 layers are partitioned similarly to MM-GIDNO ViT-Large as [2,2,18,2]. While the model only uses the last layer of the backbone feature $\mathcal{B}_4$, we have plotted intermediate features $\mathcal{B}_1, \mathcal{B}_2, \mathcal{B}_3$, and $\mathcal{B}_4$ for fair comparison. Neck in this model is actually part of Spatial pyramid transformer structure with $\mathcal{N}_1 = \mathcal{H}_1(\mathcal{C}_1(\mathcal{B}_4)) \parallel \mathcal{N}_2 = \mathcal{H}_2(\mathcal{C}_2(\mathcal{B}_4)) \parallel \mathcal{N}_3 = \mathcal{H}_3(\mathcal{C}_3(\mathcal{B}_4)) \parallel \mathcal{N}_4 = MaxPool(\mathcal{H}_3(\mathcal{C}_3(\mathcal{B}_4)))$ creating 4 neck features, called 'p3', 'p4', 'p5', and 'p6', in the original code. Similar to MMGIDNO there is an encoder-decoder structure in fusion network, with "early fusion". We only use encoder to show effect of fusion, generating generating '$\mathcal{F}$' features as $\mathcal{F}_1, \mathcal{F}_2, \mathcal{F}_3, \mathcal{F}_4$ for $\mathcal{N}_1, \mathcal{N}_2, \mathcal{N}_3, \mathcal{N}_4$, respectively. For plotting, we use $\mathcal{B}_1, \mathcal{B}_2, \mathcal{B}_3, \mathcal{B}_4, \mathcal{N}_4, \& \mathcal{F}_4$.

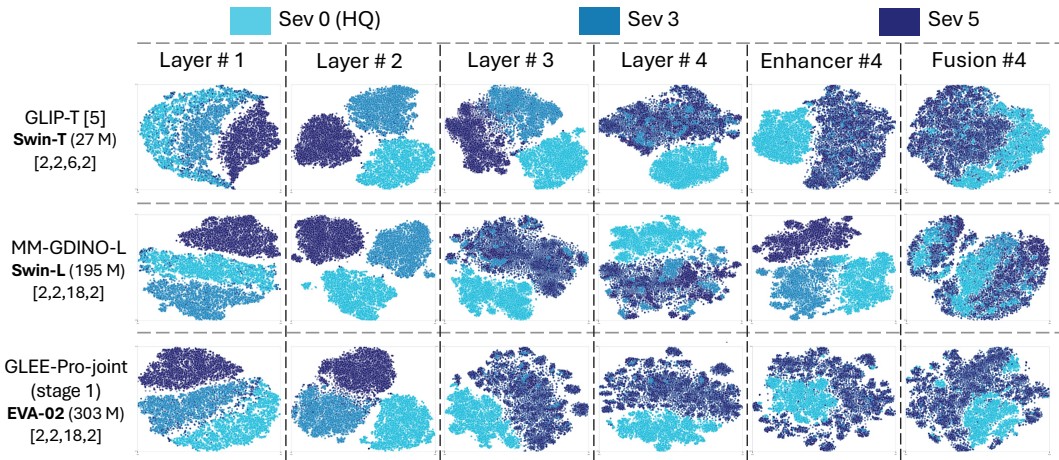

Figure 29: **Pixelation Features t-SNE:** Same details as those of Figure 8.

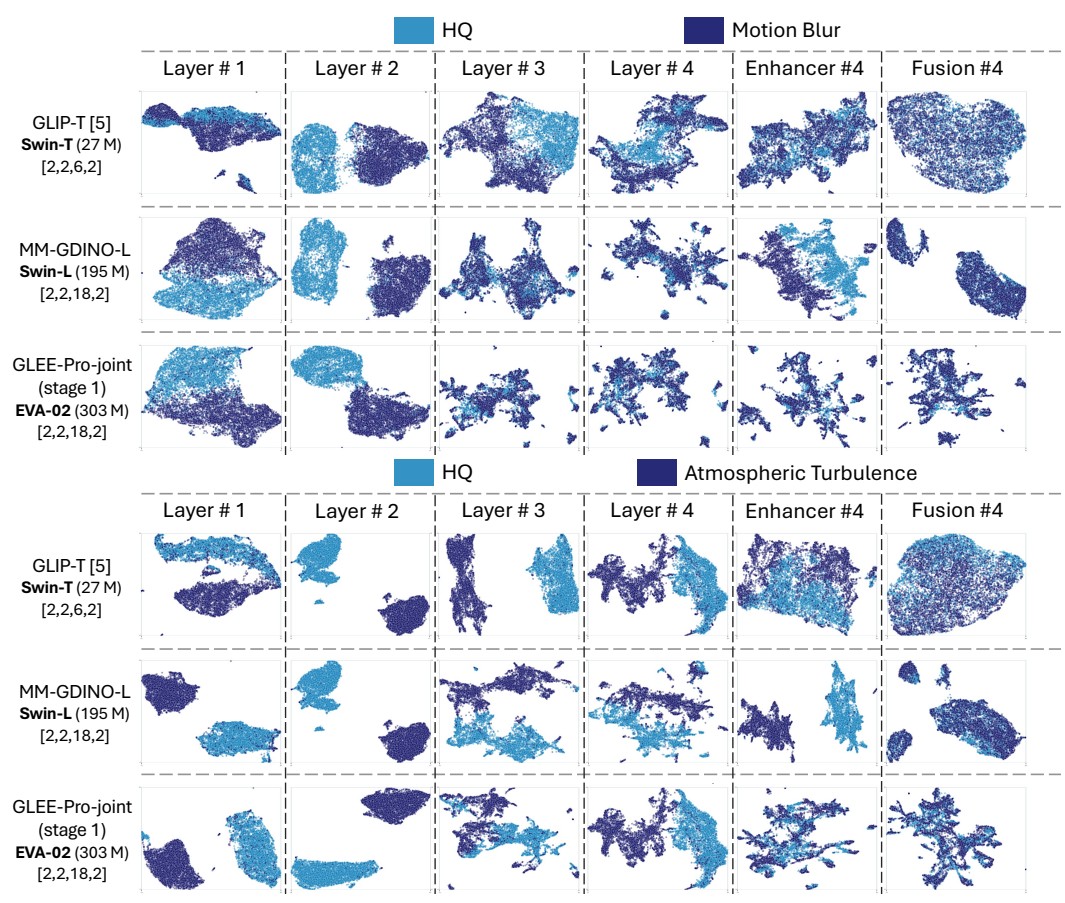

Figure 30: **Features UMAP:** Same details as those of Figure 8, for motion blur (above) and atmospheric turbulence (bottom), with noise implemented on COCO. Only 1 severity.

### D.9 FIGURE 9A

Figure 9a showed results for COCO at sev 3. Here we show results at sev 5 as well in Figure 33 and results for real world noises in Figure 32. The size of objects was determined by official annotation in COCO. The objects are categorized into three size bins—small, medium, and large—based on the area of their bounding boxes in $pixels^2$. These bins are defined as: small for areas in the range $(0, 32^2]$, medium for $(32^2, 96^2]$, and large for $(96^2, (1e^5)^2]$.

### D.10 FIGURE 9B

Figure 9b showed results for COCO at sev 3. Here we show results at sev 4 and sev 5 in (fig. 35) and results for real world noises in Figure 34. The image was divided by the number of ground truth boxes per image. Buckets with number of images $> 20$ were retained. After applying the filter, we got around 39 buckets. For each visualization, we have shown only the odd-number bucket after the 10th bucket.

### D.11 FIGURE 9C

Figure 9c showed results for COCO at sev 3. Here we show results at sev 4 and sev 5 in fig. 37 and real world noises in Figure 36. Images was divided by the summations of IOUS per image. The images are binned based upon various occlusion IOU ranges as mentioned in Fig 9c. The normalized

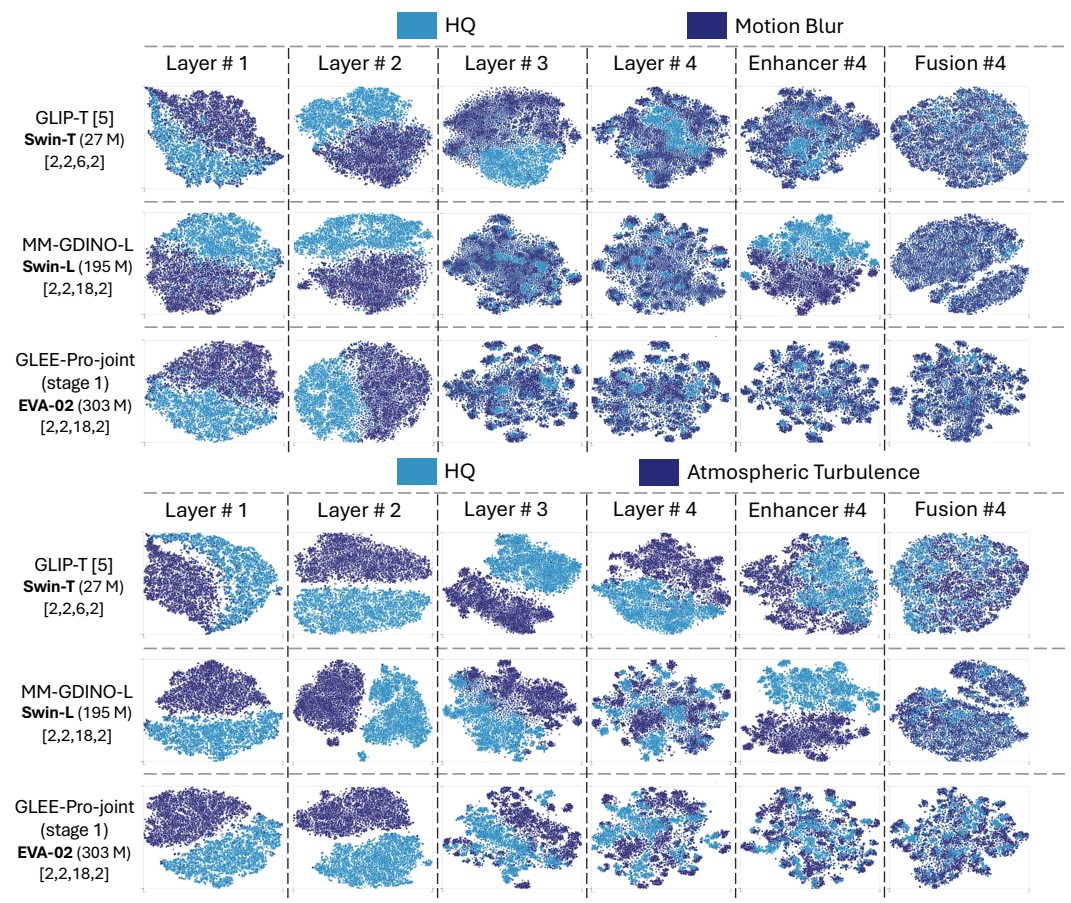

Figure 31: **Features t-SNE:** Same details as those of Figure 8, for motion blur (above) and atmospheric turbulence (bottom), with noise implemented on COCO, only 1 severity.

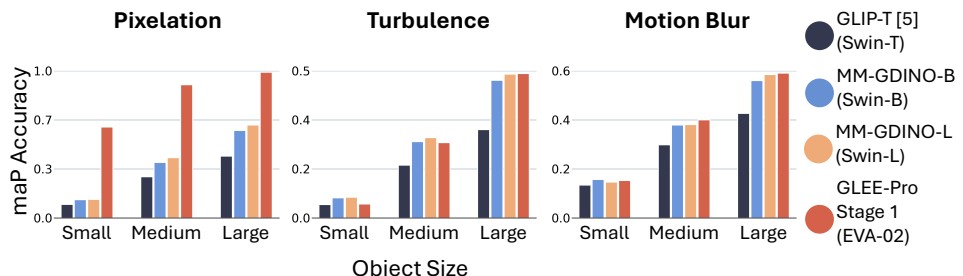

Figure 32: **Robustness vs Object Size for all perturbations**. Same details as that of Figure 9a.

per-image occlusion IOU are computed as follows:

$$\text{IoU}_{\text{image}} = \frac{\sum\limits_{(i,j)\in\mathcal{O}} \text{IoU}(B_i, B_j)}{\left| \bigcup\limits_{(i,j)\in\mathcal{O}} B_i \cup B_j \right|} \quad (1)$$

where, $\mathcal{O}$ is the set of all pairs of overlapping bounding boxes $(B_i, B_j)$. Further, the IOU bins with fewer than 50 images are removed to reduce the noise during the robustness evaluation process.

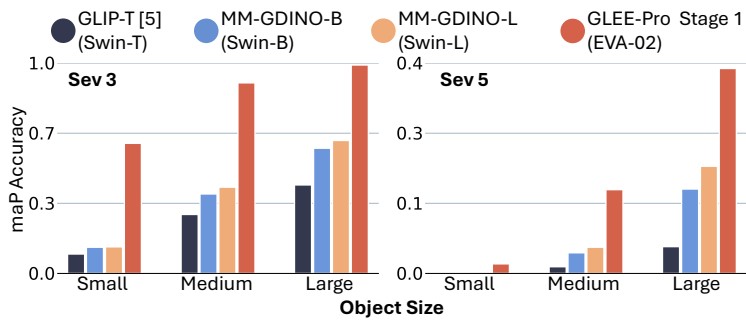

Figure 33: **Robustness vs Object Size for pixelation at sev 3 & 5**. Same details as that of Figure 9a.

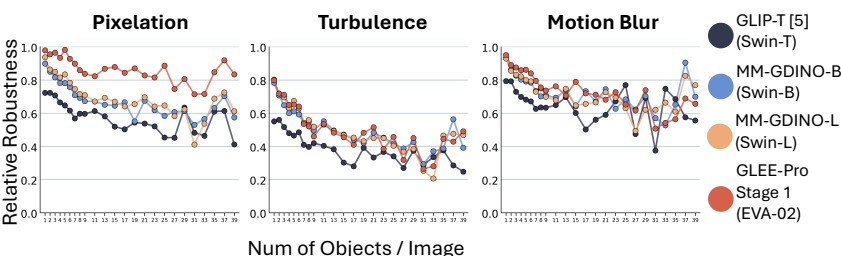

Figure 34: **Robustness vs num of objects/image for all noise perturbations**. Same details as those of Figure 9b.

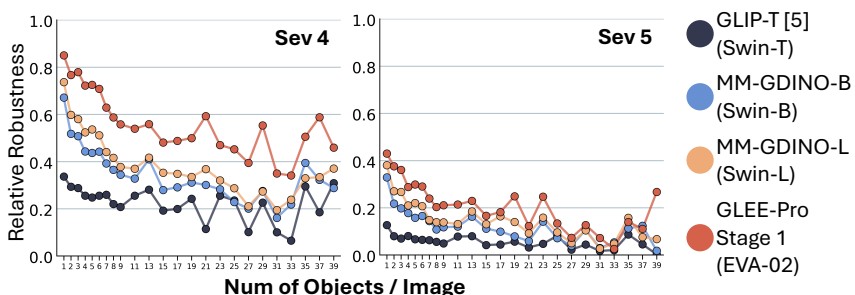

Figure 35: **Robustness vs num of objects/image at sev 4 and 5**. Same details as those of Figure 9b.

Buckets with number of images ¿ 50 were kept. After applying the filter, we got 5 bins, which are shown on X-axis.

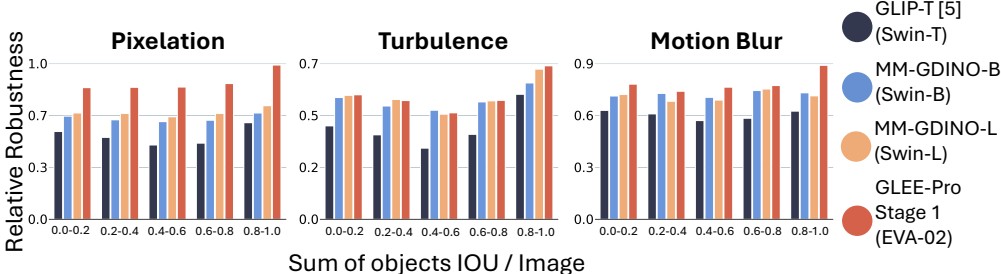

Figure 36: **Robustness vs occlusion with real world perturbation on COCO**. Same details as that of Figure 9c.

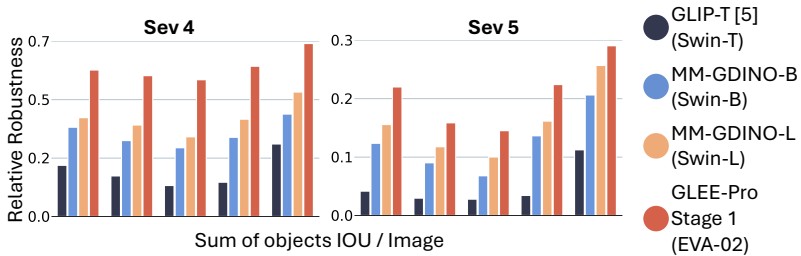

Figure 37: **Robustness vs occlusion Sev 4 for COCO**. Same details as that of Figure 9c.

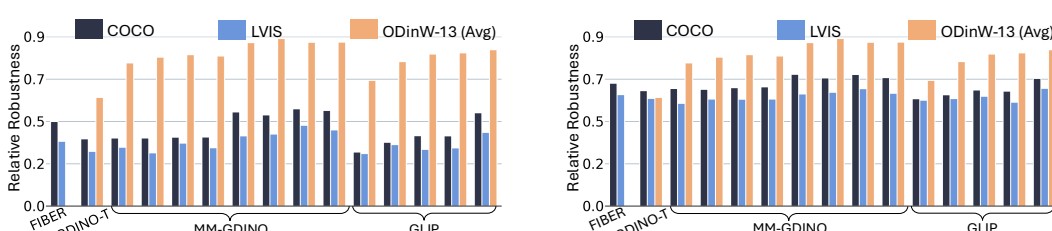

Figure 38: **Robustness of dataset under real world perturbations** (left) Robustness of dataset under turbulence perturbation (right) Robustness of dataset under motion blur perturbation. Same details as that of Section 4.3

### D.12    FIGURE 10 (LEFT)

Even with real world perturbations, the ODinW-13 maintains high robustness scores of $\simeq 0.7$ across models. However, in motion blur the overall robustness is higher across models over all the datasets due to minimal perturbation effect of motion blur.

### D.13    FIGURE 10 (RIGHT)

Section 4.3 (right) showed results for COCO classes at sev 3. Here we consider robustness vs. class for ODinW-13. For each category, we have three metrics: 1) accuracy of GLIP-T model, 2) average size of the objects (computed via mean IOU), 3) frequency of classes (# of times certain classes appear). We first cluster the categories based on the log of frequency of classes (the values indicated are the log range in each cluster). For COCO, we cluster with K=6, while for ODinW-13 clustering size is 10. Color indicates the robustness, a darker shade indicates higher robustness. The size of nodes indicates the mean IOU. We have only shown classes with # of instances for that category >100 for COCO and >10 for ODinW-13. Additionally, a lot of OdinW-13 classes overlap across classes, hence, we chose only the first occurrence of such classes.

### D.14    ROBUSTNESS ANALYSIS WITH SWIN-L BACKBONE

Figure 40 shows robustness is consistent across GLIP, MM-GDINO, and GLEE models with the Swin-L backbone. For pixelation, the maximum difference in robustness for COCO is 0.15 at sev 3, 0.09 at sev 4 & 5, while for LVIS, the differences are 0.08 at sev 3, 0.09 at sev 4, and 0.06 at sev 5. These values imply that if the model shares a similar backbone, other bells and whistles (*e.g.* modules, training strategy, and losses *etc.*) play a minimal role in increasing robustness, namely 1) Decoder: MM-GDINO and GLEE have an encoder-decoder architecture, while GLIP doesn't (only encoder). 2) Pretraining dataset: Already established in fig. 7a. 3) Pretraining strategy: GLEE's three stages of pre-training (stage 1) and stage 2-3 finetuning has minimal impact on robustness on large EVA-02 backbone, all showing similar robustness for sev 3. 4) Training losses: Different losses in GLEE, GLIP, and MM-GDINO.

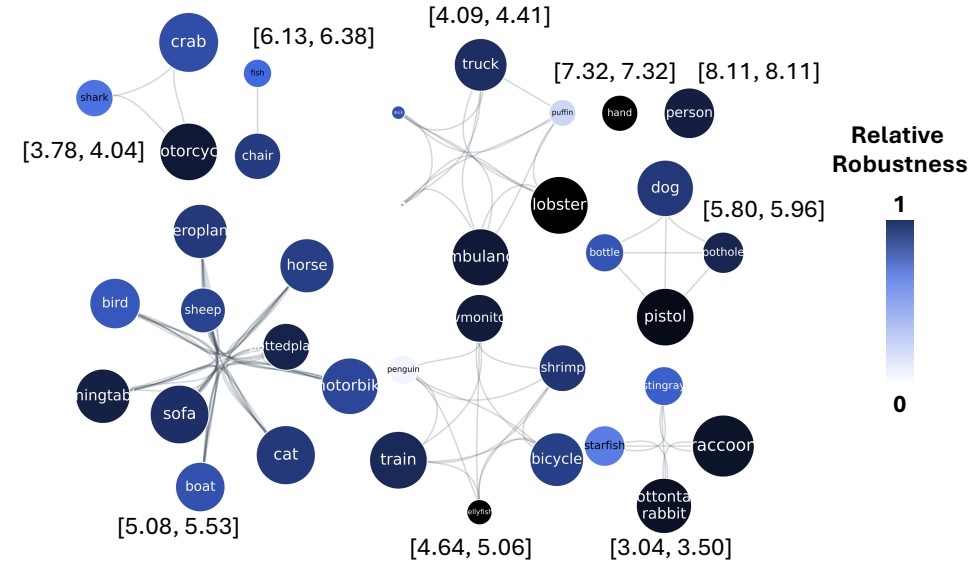

Figure 39: **Robustness vs categories for ODinW-13 at sev 3**. Same details as that of Figure 10 (right).

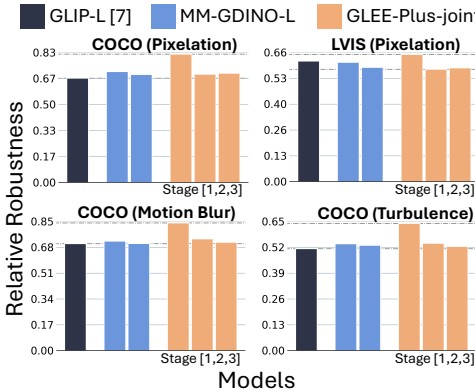

Figure 40: **Swin-L backbone consistent performance across models (Sev 3)**. Different modules across models (e.g. loss, pertaining etc.) have minimal impact on robustness if the backbone is similar. Pattern on COCO is consistent on all noises (impact of the domain of images).

# E    DATASET ANALYSIS

## E.1    COCO / LVIS VS ODINW-13

This section presents a detailed comparison between the COCO / LVIS and ODinW-13 datasets, highlighting key differences between the datasets that might be contributing to the difference between performances in the model.

**Object Size Distribution:** As shown in Figure 9a, LVIS has approximately 60% of objects below $32^2$ pixels compared to 40% in COCO. Small objects lose distinguishing features rapidly when we perturb the images.

**Spatial Characteristics:** LVIS exhibits higher object density, occlusion rates (Figure 9c), and boundary complexity, exacerbating feature ambiguity at lower resolutions.

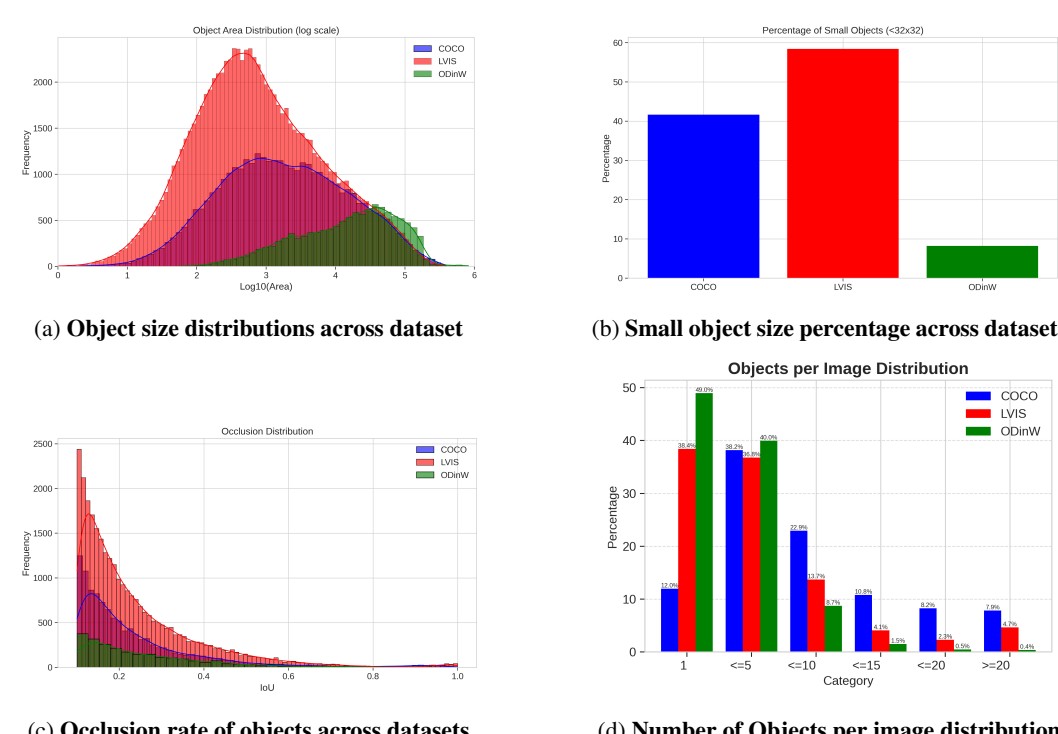

(a) **Object size distributions across dataset**

(b) **Small object size percentage across datasets**

(c) **Occlusion rate of objects across datasets**

(d) **Number of Objects per image distribution**

Figure 41: (a) LVIS shows a higher proportion of small objects compared to COCO, contributing to its greater vulnerability to resolution degradation. On the other hand ODinW-13 has much larger objects. (b) Number of small objects are more common in LVIS dataset and least common in ODinW-13 dataset. (c) Occlusion patterns reveals denser objects per image in LVIS, lowering the detection performance overall. (d) Shows the distribution of number of objects per image across dataset. ODinW-13 has the least number of objects per image across all the images.

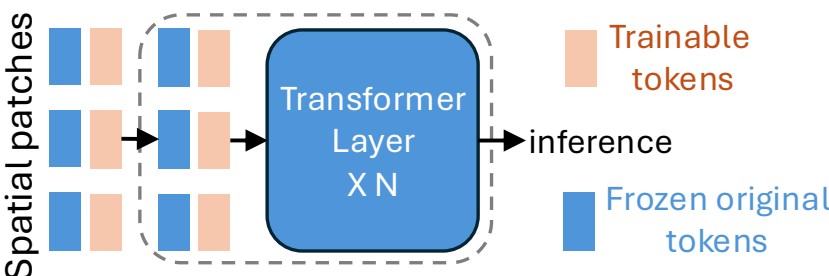

Figure 42: **LrTKO+** Trainable prompts added at every frozen layer of transformer.

## F   CURRICULUM LEARNING

### F.1   CURRICULUM LEARNING: FOREGROUND PERTURBATION (LRTK0++)

Previous works (Jarca et al., 2024; Cui et al., 2022; Kong et al., 2023; Saadabadi et al., 2024) have used Curriculum Learning to improve robustness against low resolution as an alternative to random augmentation. Curriculum Learning (Bengio et al., 2009; Hacohen & Weinshall, 2019) refers to the technique of training models, where models are slowly introduced to an increasing difficulty level. We adapt this training methodology, where the model starts with high-quality data (sev 0, easy) and gradually introduces more challenging pixelated images (sev $\in [1, 2, 3, 4]$) as training progresses. Models incrementally learn to adapt to new pixelated data distributions.

However, the general curriculum learning is not catered for object detection specificity. Hence, we progressively introduce pixelation only within the ground truth (GT) Box regions. This helps the model learn to detect pixelated objects (foreground) against the *'familiar'* high-quality surrounding (background, region outside GT box). For the first threshold $T_1 = 10^{th}$ epochs, we randomly apply the pixelation perturbation *only* within the GT box, with a probability linearly increasing from [0,1] from $0^{th} - T_1^{th}$ epochs. After the threshold $T_1$ epochs, we randomly perturb all regions outside the GT Box, progressively increasing the probability of perturbation from [0,1] until the final epoch, while GT box is sampled from severity $\in [1, 2, 3, 4]$ with a probability of 1.

### F.2 BACKGROUND PERTURBATION

Similar to Foreground Perturbation discussed in the previous section F.1, we replace the order of perturbation. Until the first threshold $T_1 = 10^{th}$ epochs, we randomly apply the pixelation perturbation region *outside* the GT box, with a probability linearly increasing from [0,1] from $0^{th} - T_1^{th}$ epochs. After the threshold $T_1$ epochs, we randomly perturb all regions inside the GT Box, progressively increasing the probability of perturbation from [0,1] until the final epoch. while the region outside the GT box is sampled from severity $\in [1, 2, 3, 4]$ with a probability of 1

### F.3 RANDOM PERTURBATION

As training progresses, the number of patches per image grows proportionally with the training epoch, reaching up to 50 patches by the final epoch. Additionally, the size range of each patch expands linearly over time, with the height and width are randomly selected from the range $(0, \min(H, W) \cdot [epoch_{current}/epoch_{total}])$. Following the approach described in section F.1, all regions within these patches are randomly perturbed. The likelihood of perturbation also increases progressively throughout training, from 0 to 1. For each patch, the severity of perturbation is from severity $\in [1, 2, 3, 4]$.

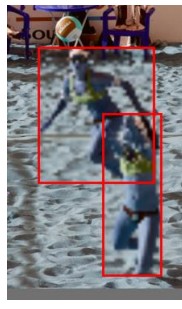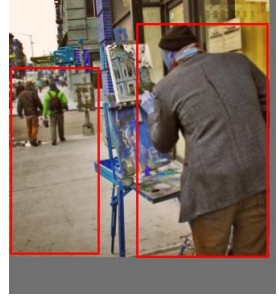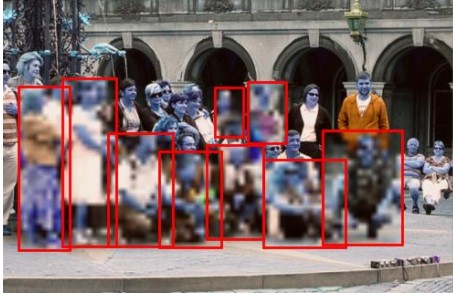

Figure 43: **Foreground Continual Learning RGB example**. Epochs 1-10 (T1) only foreground Ground-truth bounding boxes are blurred. Image taken from Flickr30k Entities

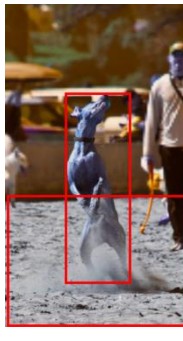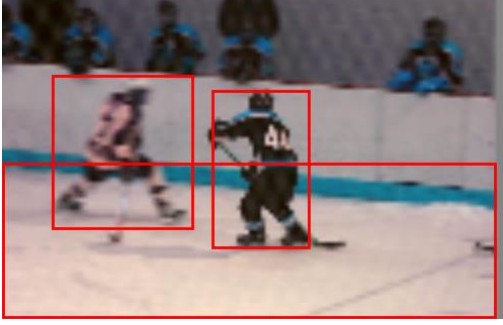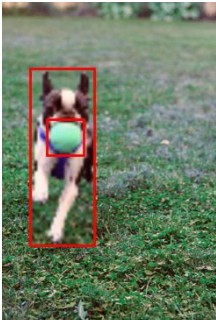

Figure 44: **Foreground Continual Learning RGB example**. Epochs 10 (T1) - 30 (max epochs) Both foreground & Backgroud gets blurred. Image taken from Flickr30k Entities

# G ETHICAL CONSIDERATIONS & LIMITATION

The diagnostic nature of our study does not obviate its ethical ramifications. We summarize the principal concerns and corresponding mitigation strategies below.

**Dual use and surveillance amplification.** Improved robustness to real-world noises can strengthen downstream systems deployed in closed-circuit television (CCTV), remote sensing, or mobile and aerial surveillance. While valuable for public-safety tasks (e.g., disaster response, wildlife monitoring), the same capability reduces the technical barrier to pervasive or covert tracking. Practitioners should adopt privacy-preserving measures and obtain explicit consent before deployment.

**Bias propagation under domain shift.** Robustness is correlated with object scale and scene composition (Sec.4.4). Small or cluttered objects exhibit sharper accuracy degradation, risking the entrenchment of dataset biases. In safety-critical contexts (autonomous driving, assistive vision), missed detections of minority classes may exacerbate inequities. Future work should couple robustness evaluation with disaggregated fairness audits spanning demographic, geographic, and socio-economic strata.

**Environmental footprint.** We demonstrate that larger transformer backbones (e.g., EVA-02) confer superior robustness. However, training and inference at this scale incur substantial energy and carbon costs. We argue for future works to explore parameter-efficient robustness techniques—such as targeted fine-tuning or curriculum learning on degraded inputs—to balance ethical imperatives of performance and sustainability.carbon emissions in line with emerging standards.

**Limitations and future safeguards.** Our scope is restricted to inference-side analysis at the moment. Based on learning that cross-exchanging information across the backbone layers can potentially help robustness remains limited to visualization of test time features. Designing a novel architecture with this kind of feature enhancer is out of the scope of the resource at hand.

Additionally, the current analysis simulates noise in a synthetic environment. We advocate an expanded robustness-and-ethics benchmark that integrates fairness diagnostics, privacy-leakage assays, and real footage, all collected under informed consent.

By foregrounding these issues, we aim to ensure that advances in robust zero-shot detection progress hand-in-hand with proactive mitigation of societal risks.

