# OpenReview forum: "Robust onion: Peeling Open Vocab Object Detectors Under Noise"
_ICLR.cc/2026/Conference — Submitted to ICLR 2026_

### Official Review · Reviewer_cRjy · 2025-10-28

**Soundness:** 2
**Presentation:** 3
**Contribution:** 2
**Rating:** 2
**Confidence:** 3

**Summary:**

This paper investigates the robustness of Open-Vocabulary Object Detectors (OVODs) under real-world noise and visual degradation.
The authors introduce Robust Onion, a systematic framework that “peels apart” different OVOD components to analyze their resilience to noise using controlled synthetic distortions. Through empirical analysis across multiple architectures and datasets, the paper finds that:

1. robustness is driven mainly by the image domain rather than annotations.

2. similar backbones exhibit comparable robustness due to shared feature collapse patterns.

3. pretraining details and captions contribute little to noise robustness.

4. common benchmarks such as ODinW-13 may give a misleading impression of robustness.

These insights highlight the need for new strategies such as cross-layer feature exchange or continual learning for building noise-tolerant OVODs.

**Strengths:**

- Comprehensive evaluation across multiple models and datasets under diverse visual distortions.

- Clear empirical dissection of factors (architecture, pretraining, annotations) affecting robustness.

- Rich analysis with quantitative and qualitative visualization results.

- Identifies key limitations of current benchmarks (e.g., ODinW-13) and provides valuable diagnostic insights.

**Weaknesses:**

- Lacks quantitative comparison against recent SOTA robust or noise-aware OVD methods.

- The motivation for studying robustness under visual noise could be better connected to real-world deployment scenarios.

- Analysis-heavy paper without a concrete methodological contribution or design proposal.

- No clear theoretical or mathematical formulation for the proposed “robust design direction.”

- Missing comparison with input-level denoising or data augmentation baselines.

**Questions:**

- How does Robust Onion compare quantitatively to recent noise-aware or robust OVD baselines beyond ODinW-13?

- What are the key real-world deployment scenarios where robustness under visual noise is most critical?

- Can the insights from this analysis lead to a concrete training or architectural strategy for improving OVD robustness?

- How does the proposed analysis differ in impact from simpler input-level methods such as denoising or augmentation?

---

> ### Author Response · Authors · 2025-11-19
> **Rebuttal - 1**
>
> Hello Reviewer cRjy. Thank you so much for the helpful feedback. We have addressed your questions and comments below. To better answer the queries, we have responded to weaknesses and questions jointly.
>
> ---
> ---
> ```
> W1. Comparison against SOTA
> ```
> Our baseline is the most recently published robustness technique, LR-TK0 (ICLR 2025, [6]), SOTA in VLM-based image classification. For OV-OD, there are no training methods / techniques that adapt models for robustness to the best of our knowledge. Other published works [7, 9] improve robustness by leveraging pretrained weights, with [7] training VLM classifiers on the test domain (defeating the purpose of our analysis) and [9] adapting image segmentation SAM.  LR-TK0 was shown to outperform Super Resolution (6 SR methods), RobustSAM (CVPR 2024 [9]),  and VPT (ECCV 2022 [11]). Hence, we chose [6] as our baseline to verify our analysis, `Table 1 & Table 2`.
>
> ---
> ---
> ```
> W2. & Q2. Real-world deployment.
> ```
>
> We appreciate reviewers' suggestions to strengthen our motivational framing and will add the following motivation via connections to the real world in the revised submission (camera-ready extra 10th page).
> Our study is directly motivated by the "zero-shot” / “generalized” use of open-vocabulary object detectors in the real-world, especially in high-stakes, unconstrained environments where visual degradation is intrinsic and re-training is not practical and often impossible.
> Our analysis targets critical domains where degradation is a primary operational challenge:
> -  *Autonomous Driving:* Models must handle **motion blur** (from vehicle/pedestrian movement), **atmospheric turbulence** (hot air distortion), and weather (fog, rain, snow). Crucially, detecting distant objects like pedestrians or traffic signals, vital for safe braking distances, is a long-range detection problem frequented by severe **pixelation**.
> - *Security and Surveillance (CCTV)*: This domain is populated by low-resolution cameras. Robustness to **pixelation** and sensor noise is not an edge case but the default operational condition.
> - *Consumer Photography & Video*: AI models in smartphones must process images with **motion blur** (hand shake), **pixelation** (digital zoom), and JPG compression artifacts. Likewise, models analyzing video frames on streaming platforms must constantly handle **motion blur**.
> - *Satellite Imagery*: Analysis of astronomical or earth-observation imagery often comes hand in hand with the problem of extreme **pixelation**.
>
> To ensure our findings are grounded, the choice of noises was not arbitrary. As detailed in `Fig 2, Line 071 & Section 3`, including some recent studies [6,7,8,10], noise is essentially feature / variance collapse, despite its appearance in RGB space. We show this via the real-world self-driving car BDDK100 dataset, where Real-world Cloudy, Overcast feature collapse is similar to synthetic Turbulence, Rain, Snow, and ISO. Similarly, real-world Snow, Rain is similar to Motion Blur, JPG, Fog, and Focus Blur. Among these two categories, we have chosen one synthetic noise from each to analyze i.e., **Turbulence and Motion Blur**.
>
> Finally, we took a step further in synthetic simulation by validating our analysis on the Wider Face dataset in `Section 5 (LR-TK0++)` , demonstrating that our insights regarding backbone robustness can translate directly to improving performance on naturally occurring, uncontrolled noise in the wild.

---

> ### Author Response · Authors · 2025-11-19
> **Rebuttal - 2**
>
> ```
> W3. Concrete methodological / design proposal.
> &
> Q3. Concrete training or architectural strategy
> ```
>
> Thank you for the question. Our `Section 5` LrTK0+ & LrTK0++ validate some of the insights from our analysis in `Section 4.1 and Section 4.2`, i.e., *improving the robustness of the entire object detector (GLIP), without fine-tuning on COCO or ODinW-13 or Wider face dataset*, via:
> - Trainable tokens in **only on the visual backbone**, especially in shallow layers
> - **No modifications to any other architectural components** (5 modules (3 transformers), fusion network, enhancer network, text-vision feature alignment layer, bounding box generation, and classifier).
> - **No modifications to the Language** backbone / text features
>
> Our **[Takeaways & Model Design]** lists several other insights which we believe will be benficial in designing future robust object detectors (training & architectural strategy):
>
> - `[Line 313]` Cross-exchange of information between vision layers should help impart robustness across layers
> - `[Line 269]` Given resource constraint environments, Swin-B can be an amazing alternative to EVA-02 (4x bigger), and Swin-L (2x bigger). with similar robustness.
> - `[Line 290]` Model design should incorporate explicit noise-robust training, without relying on pretraining and fine-tuning
> - `[Line 369]` Datasets like ODinW-13 ( large, singular objects) can overstate robustness, without reflecting models’ true robustness.
> - `[Line 371]` Robustness seems to largely depend on diverse image domains rather than the type of annotation.
> - `[Line 416]` Expressive captions (finetuning), and prompts engineering with the context of degradations (evaluation) do not significantly improve ‘visual’ robustness.
> - `[Section 5, LR-TK0+ | LR-TK0++]` Continual learning focusing robustness of shallow layers with minimal trainable parameters (~5.7% additional parameters) helps improve robustness in a resource-constrained environment where Distillation may not be feasible.
>
> Our apologies for the confusion and for not making it clearer, but Robust Onion is a detailed **“empirical” analysis** of open vocabulary object detectors (OV-ODs), as already noted by all reviewers. It's **not a proposed technique** for explicitly improving robustness. Our paper is submitted as an **“interpretability and explainable AI”** paper, with the primary goal to analyze the influence of noises on OVDs (`line 79`) & narrow down the bottlenecks, providing multiple actionable insights and not limited to the proposition of a singular solution for fixing the models (`line 95`). In simple words, we want to open up the black box of open vocabulary object detectors (OV-ODs) to see how and where noise impacts these models.
>
> ---
> ---
> ```
> W4. No theoretical or mathematical formulation
> ```
>
> We acknowledge the reviewer's issue with the lack of a theoretical/mathematical formulation in our analysis. Robust Onion is a detailed *empirical analysis of the influence of noises on open vocabulary object detectors* (OV-ODs), as already noted by all reviewers. Research in robustness, especially for heavier models such as object detectors, is vastly underexplored, with a very small handful of analyses. Our work is in line with existing research (including the ones cited by the reviewer), all are empirical with little to no theoretical formulation [1,2,3,4,5]. On the contrary, most existing research is black box benchmarking analysis, while our intends to open up the black box of (OV-ODs) to see how and where noise impacts these models.

---

> ### Author Response · Authors · 2025-11-19
> **Rebuttal - 3**
>
> ---
> ---
> ```
> W5. Missing comparison
> &
> Q4. Denoising or augmentation?
> ```
> Thank you for the question, and apologies for not clarifying it in our paper. `Table 1` lists multiple variants of data augmentations, including our LR-TK0++, which itself is a data augmentation technique. Our proposed possible research direction (LR-TK0+/TK0++) is just a **“Validation of our analysis”** (Section 5 is renamed, apologies for the confusion). Our analysis doesn't really overlap with strategies for fixing noise.
>
> Our baseline (recently published prompt-based LR-TK0 (ICLR 2025, [6])) to verify our analysis, `Table 1 & Table 2` was shown to outperform Super Resolution (6 SR methods), RobustSAM (CVPR 2024 [9]),  and VPT (ECCV 2022 [11]). Denoising (super resolution) / augmentation, with key differences with our analysis include:
>
> - *Denoising/augmentation is noise-specific*: There is no universal super-resolution that can fix all noises. While the observations we are making for noises are regardless of the noises, encompassing turbulence, pixelation, and motion blur. These noises mimic the feature collapse of the real world.
>
> - *Denoising / Augmentations approaches are model agnostic*: They act on the input image and do not really reveal anything about how modified images impact object detectors. Our analysis, on the other hand, highlights how noisy features are dependent on model depth (`Section 4.2`), how input image impacts robustness (`Section 4.3`), and how language impacts robustness (`Section 4.4`).
>
> - *Denoiser rarely works in a zero-shot setting (a model that can remove noise from any image of any object)*: Lr-tk0 [6] showed how 6 super-resolution models fail under the unseen category of pixelated “cat”. While we have zero-shot analysis of COCO / LVIS, ODinW-13, RefCOCO/RefCOCOg/RefCOCO+, and Flickr30k.
>
> ---
> ---
> ```
> Q1. Beyond ODinW-13?
> ```
> Yes, We have shown LR-TK0+ and LR-TK0++ improvements on COCO (`Table 1 & 2`), ODinW-13 (`Table 1`), and Wider face  (`Table 3`). We would also be happy to clarify that Robust Onion is a detailed empirical analysis of OVD models on (`Line 131`):
> - **COCO & LVIS** Analysis (Most common Object Detection dataset)
> - **ODinW-13** for zero-shot validation of our analysis and dataset-based analysis
> - **RefCOCO (2014), RefCOCO+ (2016), RefCOCOg (2016)**, for language-based analysis
> - **Flickr30k** for language-based analysis and training models
> - **Wider Face** for zero-shot evaluation
>
> Our work (“interpretability and explainable AI”) primarily analyzes the influence of noise on OVDs and is not limited to the proposition of a singular solution for fixing the models. Thus, comparison with other OVD baselines isn't really applicable here.
>
> ---
> ---
>
> Please accept our sincere apologies for this long rebuttal, with some repetitions. We hope to have answered all the weaknesses and questions. We would like to thank you once again for the feedback, and we will be happy to answer additional questions/concerns.
>
> ---
> ---
> References
> [1] Open-Vocabulary Object Detectors: Robustness Challenges under Distribution Shifts, ECCV, 2024
> [2] Benchmarking object detection robustness against real-world corruptions, IJCV 2024
> [3] Coco-o: A benchmark for object detectors under natural distribution shifts, CVPR 2023
> [4] The Effects of Super-Resolution on Object Detection Performance in Satellite Imagery. CVPR 2019
> [5] Denet: Detection-driven enhancement network for object detection under adverse weather conditions, ACCV 2022
> [6] LR0.FM: Low-Res Benchmark and Improving Robustness for Zero-Shot Classification in Foundation Models. ICLR 2025
> [7] Mint: A Simple Test-Time Adaptation of Vision-Language Models against Common Corruptions, Neurips 2025
> [8] Dive into the Resolution Augmentations and Metrics in Low Resolution Face Recognition: A Plain yet Effective New Baseline. AAAI 2023
> [9] Robustsam: Segment anything robustly on degraded images. CVPR 204
> [10] Recognizability Embedding Enhancement for Very Low-Resolution Face Recognition and Quality Estimation, CVPR 2023.
> [11] Visual prompt tuning. ECCV 2022

---

### Official Review · Reviewer_gY5W · 2025-10-30

**Soundness:** 3
**Presentation:** 2
**Contribution:** 3
**Rating:** 4
**Confidence:** 4

**Summary:**

This paper presents robustness analysis of open-vocabulary object detectors (OV-ODs) under common visual corruptions like pixelation, motion blur, and turbulence. The authors propose the Robust Onion framework, which isolates how different model components contribute to robustness. Evaluating six models (e.g., GLIP, MM-GDINO, GLEE) across COCO, LVIS, ODinW-13, and Wider Face, the study finds that backbone depth, not fine-tuning or caption supervision, dominates robustness behavior. It also introduces two lightweight continual learning strategies (LR-TK0+ and LR-TK0++) to improve robustness in zero-shot settings.

**Strengths:**

S1. The paper introduces a clear analytical framework to “peel” away layers of complexity and pinpoint which components of OV-OD models contribute to or detract from robustness. By systematically turning off or swapping certain components (e.g. using a frozen vs. fine-tuned backbone, or evaluating with vs. without caption-based training), the authors can attribute robustness (or lack thereof) to specific factors. This level of analysis moves beyond treating the model as a black box – it provides insightful breakdowns of how different stages (backbone, detector head, multimodal fusion, etc.) behave under noise.

S2. The study delivers some important insights. One of them, the discovery that backbone depth/capacity is the primary driver of robustness (more so than fine-tuning or the richness of text supervision) is a interesting fact for further work.

S3. The paper makes a practical contribution by proposing simple fine-tuning strategies (LR-TK0+ and LR-TK0++) that yield noticeable robustness gains. These strategies are lightweight (avoiding full model retraining) and thus would be attractive for practitioners looking to harden existing OV detectors against noise. The fact that a gradual fine-tuning on noisy data (LR-TK0++) outperforms a naive augmentation approach is a useful takeaway for the field.

S4. Writing and structuring of the paper is easy to follow.

**Weaknesses:**

W1. Contribution - While the analysis is detailed, the paper’s contributions are primarily empirical. The lack of a strong algorithmic or theoretical innovation prevent the exact take home knowledge to advance the field of OVOD. The proposed robustness fixes (LR-TK0+/TK0++) are relatively simple fine-tuning heuristics rather than fundamentally new methods which means, the contributions in applied and theoretical research are limited.

W2.  Comparisons with Prior Work: The paper does not explicitly situate itself against closely related robustness studies. For instance, Chhipa et al. (2024) [1] evaluates open-vocabulary detectors (OWL-ViT, YOLO-CLIP, Grounding DINO) under distribution shifts and corruptions finding significant performance drops as well. Similarly, in the broader object detection literature, there have been benchmarks for robustness to corruptions (e.g. COCO-C and BDD100K-C) introduced by Liu et al. (2024) [2]. These works revealed, for example, that even high-mAP detectors can be very brittle and that transformer-based detectors may handle corruptions better than older architecture. It is suggested to have combined reasonable analysis with such studies and provide clear insights your findings.

[1]  Chhipa, Prakash Chandra, et al. "Open-Vocabulary Object Detectors: Robustness Challenges Under Distribution Shifts." European Conference on Computer Vision. Cham: Springer Nature Switzerland, 2024

[2] Liu, Jiawei, et al. "Benchmarking object detection robustness against real-world corruptions." International Journal of Computer Vision 132.10 (2024): 4398-4416.

W3. This paper propose fine-tuning strategies (LR-TK0 series) but do not compare them to alternative robustness interventions like standard data augmentation training or adversarial training. It’s mentioned that LR-TK0++ beats “random augmentation” (presumably LR-TK0+), but a stronger baseline could be full data augmentation during initial training (for instance, training the detector on corrupted images from scratch or heavy augmentation schedules). Evaluating such a baseline would show how far the proposed lightweight approach is from what one could achieve with more extensive retraining.

W4. The claim that backbone depth alone drives robustness might be somewhat oversimplified. There is a correlation in their results, but correlation does not guarantee causation. Deeper models often also differ in other aspects: e.g. architecture family (CNN vs Transformer), pre-training dataset size, or training strategies. It’s possible that the robustness comes from some of these factors (for example, transformer-based detectors might inherently be more robust to certain perturbation. I suggest to provide convincing argument for that.

W5. Scope of Noise Types: The study covers three synthetic noise types (pixelation, gaussian blur, turbulence). These mainly represent low-level distortions blurring or obscuring the image. While these are important, the robustness problem has other dimensions that the paper does not address – for example, illumination changes, weather effects (rain, fog).

**Questions:**

Please refer weakness section. I can change the score based on rebuttal's response.

---

> ### Author Response · Authors · 2025-11-18
> **Rebuttal - 1**
>
> We thank the reviewer for the constructive feedback and for recognizing the value of our "Robust Onion" framework (S1) and the practicality of our proposed solutions (S3). We address the concerns below with specific technical evidence from our analysis.
>
> ---
> ---
> ```
> W1. Limited contributions in applied and theoretical research.
> ```
> We acknowledge the reviewer's issue with the lack of a theoretical / mathematical formulation in our analysis. To better answer this, we will break the weakness into two parts :
>
> ```
> Contribution...
> ```
> Our apologies for the confusion and not making it clearer, but Robust Onion is a **detailed empirical analysis** of open vocabulary object detectors (OV-ODs), as already noted by reviewers themselves, “analysis is detailed”. Our paper is submitted as an **“interpretability and explainable AI”** paper, with the primary goal to analyze the influence of noises on OV-ODs (line 79) & narrow down the bottlenecks, and not necessarily to propose a solution for fixing the models (line 95). In simple words, we want to open up the black box of (OV-ODs) to see how and where noise impacts these models.
> Research in robustness, especially for heavier models such as object detectors, is vastly underexplored, with a very small handful of analyses. Our work is in line with existing research (including the ones cited by the reviewer), all are empirical with little to no theoretical formulation [1,2,3,4,5].
>
> ```
> The proposed robustness fixes (LR-TK0+/TK0++)
> ```
> We sincerely apologize for the confusion caused. Our `Section 5 (LrTK0+ & LrTK0++)` is a **“validation of our analysis”** (section is renamed to reflect it), and **not a proposed technique** for explicitly improving robustness.  LrTK0+ & LrTK0++ validate our analysis in `Section 4.1 and Section 4.2`, i.e. *for improving the robustness of the entire object detector (GLIP)*, without fine-tuning on COCO or ODinW-13 or Wider face dataset, via:
> - Trainable tokens in **only on the visual backbone**, especially in shallow layers
> - **No modifications to any other architectural components** (5 modules (3 transformers), fusion network, enhancer network, text-vision feature alignment layer, bounding box generation, and classifier).
> - **No modifications** to the Language backbone / text features
>
> ---
> ---
> ```
> W3. Missing Comparison
> ```
> Thank you for the question, and apologies for not clarifying it in our paper. `Table 1` lists multiple variants of data augmentations, including our LR-TK0++, which itself is a data augmentation technique. Our proposed possible research direction (LR-TK0+/TK0++) is just a **“Validation of our analysis”** (Section 5 is renamed, apologies for the confusion), and **not a proposed solution**. Hence, comparison with other baselines may not be applicable here.
> To answer this weakness properly, we would split the query into components :
> ```
> Extensive retraining
> ```
> VLMs (e.g. CLIP, EVA) are trained on **400 Million - 2 Billion images-text pairs** and models (e.g., GLEE trained on 18 datasets). Re-training from scratch every time a new noise appears will be non-scalable. Given the resource, it will be infeasible for us to train even the most basic GLIP model from scratch (trained on ~27 M image-text pairs, on 32 GPUs in 2 stages). The recently published works [6, 7, 9] improve robustness by leveraging pretrained weights. Hence, as our baseline, we opt for a prompt-based approach (LR-TK0) [6 ,9] that adapts on pretrained weights for robustness.
> ```
> Data augmentation/Adversarial training.
> ```
> `Table 1` lists multiple variants of data augmentations (`row 1` uniform is standard data augmentation), including our LR-TK0++, which itself is a data augmentation. Augmentation approaches are model agnostic, as they act on the input image and do not really reveal anything about how modified images impact object detectors. Our analysis, on the other hand, highlights how noisy features are dependent on model depth `(Section 4.2)`, how input image / language impacts robustness `(Section 4.3 & 4.4)`.
>
> Adversarial Robustness (Adversarial training) is robustness against adversarial attacks. While having a similar name to “robustness,” its robustness is against “attack” targeted at a specific model designed to make the predictions wrong. We are not aware of any noise-based adversarial training.
>
>
> ---
> ---
> **References in last rebuttal comment**

---

> ### Author Response · Authors · 2025-11-18
> **Rebuttal - 2**
>
> ```
> W4. The claim that backbone depth alone drives robustness
> ```
> We appreciate a deep, insightful question and will try to do our best to make our case. We first want to clear the confusion; we are claiming that the visual backbone determines the robustness (`line 267`), while other components like pretraining strategy / dataset (`Fig. 7`), architectural modules (like encoder / decoder, feature enhancer / neck, box predictor / RPN, classifier) (`Fig. 6b`), language features / input prompts (`Sec 4.4`), type of annotation of dataset play a minimal role (`line 371`). Size/depth plays a limited role for sufficiently large backbones (`line 291`). In order to successfully make our case, we would break the question down part by part.
>
> ```
> CNN vs Transformer
> ```
> Our deeper analysis centers around Transformers only, however, unless fine-tuned, ResNet-based models are generally inferior to the Transformer. Pixelation (`Fig. 3`), Turbulence & motion blur (`Fig. 19`).
>
> ```
> pre-training dataset size or training strategies
> ```
> E.g. after similar ImageNet (1M-15M) pre-training of swin transformers, backbones undergo different stages of training **~400 M - 10 Billion image-text pairs in different stages** (distillation, pretraining, fine-tuning). Models like GLEE, MM-GDINO, and GLIP use vastly different pre-training datasets and strategies (e.g., GLEE on 18 datasets in 3 stages, MM-GDINO on 9 datasets, and GLIP via distillation).
>
> Our observation (`Fig. 6b`) shows that despite these different pre-training schemes, similar backbones have similar robustness. This strongly implies that the "bells and whistles of pretraining or overall architecture" play a fairly minimal role. `Fig 7a)` shows that, despite different training dataset sizes, robustness has a very weak to no correlation with pretraining dataset or training strategies (Correlation on COCO / LVIS: 0.34 / 0.28 for pixelation, 0.37 / 0.37 for turbulence, and 0.34 / 0.23 for motion blur (`line 255`)).
> We show a similar trend that with similar backbones have similar robustness for all noises in our study (11 noises)  in the `supplementary Fig 28`. Among the 21 backbones shown, there are only 2 exceptions GLEE PLUS Joint (Stage 1), and GLIP-A, which deviate from the pattern by 0.1.
>
> ---
> ---
> ```
> W5. Scope of Noise Types
> ```
> We are thankful for the constructive advice on making our work more generalizable. One correction: we have shown the analysis for pixelation, motion blur, and turbulence. Our work (`Fig. 2`), including some recent studies [6, 7, 8, 10], has demonstrated that noise is essentially feature/variance collapse, despite its appearance in RGB space. That's the core of our analysis in `Figure 8`, explaining why similar depths have similar robustness. Establishing illumination as a feature collapse would be beyond the scope of this work.
>
> We have chosen **Turbulence** as a representative of the group exhibiting “Observable Collapse” (this includes Rain, Snow) and **Motion Blur** from the group exhibiting “Minimal Collapse” (`Fig. 2`). These groups correspond to real self-driving car datasets' noises as well, where “Observable Collapse” encompasses cloudy and overcast skies, and “Observable Collapse” encompasses Snow and Rain.
>
> One of the key takeaways of our work is
> *“​The choice of backbones decides the robustness, while other bells and whistles, like additional modules (e.g. MMGDINO/GLEE decoder), and extensive pre-training (e.g. MM-GDINO pre-trained on 9 datasets, and GLEE on 18 via three stages of training etc.) play a minimal role.”* (`line 266`).
> We showed this analysis for **all the noises in this study, including rain and fog** (supplementary Fig 28, values and shades of blue).
>
>
> **For Rain:**
> *Swin-T (MM-GDINO, GLIP) < Swin B (MM-GDINO) ~ Swin-L (MM-GDINO, GLIP, GLEE) <= EVA (GLEE)*
> Which is similar to Turbulence from the similar feature collapse group
> *Swin-T (MM-GDINO, GLIP) < Swin B (MM-GDINO) ~ Swin-L (MM-GDINO, GLIP, GLEE) ~ EVA (GLEE)*
>
> **For Snow:**
> *Swin-T (MM-GDINO, GLIP) < Swin B (MM-GDINO) ~ Swin-L (MM-GDINO, GLIP, GLEE) ~ EVA (GLEE)*
> Which is almost identical to the motion blur from the similar feature collapse group
> *Swin-T (MM-GDINO, GLIP) < Swin B (MM-GDINO) ~ Swin-L (MM-GDINO, GLIP, GLEE) ~ EVA (GLEE)*
>
> Different models have similar robustness because of similarity in backbones. And Swin-B and Swin-L performance are almost similar because of the depth of 24.
>
> Finally, we took a step further in synthetic simulation by validating our analysis on the **Wider Face dataset** in `Section 5 (LR-TK0++)`, demonstrating that our insights regarding backbone robustness can translate directly to improving performance on naturally occurring, uncontrolled noise in the wild.

---

> ### Author Response · Authors · 2025-11-18
> **Rebuttal - 3**
>
> ```
> W2. Comparisons with Prior Work:
> ```
> Thank you so much for the constructive reference. We will include the following elaboration in the final copy to differentiate our work from the existing work.
>
> **Core Key difference:**
> ```
> [1] Chhipa, Prakash Chandra, et al. (2024) "Open-Vocabulary Object Detectors: Robustness Challenges Under Distribution Shifts." European Conference on Computer Vision.
> ```
> The referred work [1] basically defines robustness as the distribution shift impact on models (via COCO-O, and COCO-DC), e.g. high-quality image of a painting, cartoon, or tattoo. While our work primarily deals with noises, which have a unique property of feature collapse (similar to COCO-C). Our work `(Fig. 2)`, including some recent studies [6, 7, 8, 10], has shown that noise is essentially feature/variance collapse, i.e., noise, despite what they look like in RGB space, is similar with respect to feature collapse. Establishing that every form of distribution shift (like painting, cartoon, or tattoo) as a feature collapse is beyond the scope of our work.
>
> ```
> [2] Liu, Jiawei, et al. "Benchmarking object detection robustness against real-world corruptions." International Journal of Computer Vision, 132 (10), 4398-4416. (2024)
> ```
> Similar differences with [2] exist: noises like illumination, black lines, and color do not necessarily mean feature collapse, and establishing it is beyond the scope of our work. Another key difference is that it's a ResNet-heavy (non-open vocabulary), they train (fine-tune) their models on COCO before analysis, while we are primarily an open vocabulary model with both zero-shot and finetuning-based robustness analysis.
>
> Overview of key differences in “empirical” insights of [1,2] include :
> | Category | [1] ECCV'24 | [2] IJCV'24   | Our (Robust Onion)
> | :-------: | :------- | :------- |  :------- |
> | **Type** | Zero-shot Benchmark on COCO Variants on **2 transformers** | Analysis (finetuned) on COCO Variants and BBDK100 for ResNet and **2 transformers**.  | Transformer-centric analysis of zero-shot & fine-tuning on COCO Variants (noise, LVIS, RefCOCO / RefCOCOg / RefCOCO+), ODinW-13, Flickr30k, Wider Face with **29 transformers variants**. |
> **Goal**  | Black-box evaluation (No unboxing) to determine which models are better under what conditions (without determining the underlying cause) | Non-open vocabulary ResNet analysis (No unboxing) focusing on training strategy | Investigation-centric (Answer “Why” / “Where” / ”How” there is a performance drop in open-vocabulary transformers).  Benchmark in setup `(Sec 3 & Fig. 5)`. Minimal focus on ResNets.
> **Takeaway [1]** | Grounding DINO exhibits the least performance drop compared to OWL-ViT, YOLO World on COCO-C | | We explain this via **performance linear relationship with robustness** (`Figure 5b)`), so “Grounding DINO (48.4) > YOLO World (39.3) > OWL-ViT (26.4)” implies robustness ordering as well.
> **Takeaway [2]** | | mAP and robustness are not linear.  | We explain this by indicating: **Similar backbones have similar robustness** (`Figure 6b)`) and DETR and Deformable DETR have the same transformer backbone.
>
> ---
> ---
> Please accept our sincere apologies for this long rebuttal, with repetitions. We would like to thank you once again for the feedback, and we will be happy to answer additional questions/concerns.
>
> ---
> ---
> **References**
> [3] Coco-o: A benchmark for object detectors under natural distribution shifts, CVPR 2023
> [4] The Effects of Super-Resolution on Object Detection Performance in Satellite Imagery. CVPR 2019
> [5] Denet: Detection-driven enhancement network for object detection under adverse weather conditions, ACCV 2022
> [6] LR0.FM: Low-Res Benchmark and Improving Robustness for Zero-Shot Classification in Foundation Models. ICLR 2025
> [7] Mint: A Simple Test-Time Adaptation of Vision-Language Models against Common Corruptions, Neurips 2025
> [8] Dive into the Resolution Augmentations and Metrics in Low Resolution Face Recognition: A Plain yet Effective New Baseline. AAAI 2023
> [9] Robustsam: Segment anything robustly on degraded images. CVPR 204
> [10] Recognizability Embedding Enhancement for Very Low-Resolution Face Recognition and Quality Estimation, CVPR 2023.

---

### Official Review · Reviewer_x5K3 · 2025-10-31

**Soundness:** 3
**Presentation:** 3
**Contribution:** 3
**Rating:** 6
**Confidence:** 3

**Summary:**

This paper presents Robust Onion, an extensive empirical analysis of Open-Vocabulary Object Detectors (OV-ODs) under visual noise and distortions. The work introduces a systematic framework to “peel apart” different components of OV-ODs (backbones, fusion modules, pretraining datasets, fine-tuning strategies, etc.) using controlled synthetic degradations that approximate real-world noise such as turbulence, motion blur, and pixelation. The authors evaluate six prominent models (GLIP, FIBER, MM-GDINO, GLEE, YOLO-World, and RegionCLIP) across multiple datasets (COCO, LVIS, ODinW-13). Key findings suggest that robustness is primarily driven by the vision backbone, particularly its depth and scale, while language features, annotations, and pretraining data contribute little. The analysis also highlights that robustness correlates with object size and domain rather than annotation type, and that prompt engineering or caption expressiveness has minimal effect. Finally, the authors propose LR-TK0+ and LR-TK0++, lightweight continual learning extensions designed to enhance robustness in zero-shot settings.

**Strengths:**

Provides a comprehensive empirical dissection of robustness factors in open-vocabulary object detection, covering multiple architectures, datasets, and controlled noise settings.

The experimental setup is methodologically clear and systematic, using synthetic degradations to emulate real-world noise with qualitative and quantitative alignment (as shown in Figures 1–3).

Offers important and actionable insights, such as the dominance of backbone features in determining robustness, limited effect of pretraining size or language inputs, and the misleading robustness impression given by ODinW-13 due to large-object bias.

Introduces lightweight continual learning strategies (LR-TK0+, LR-TK0++) that show measurable improvement on COCO and WiderFace without retraining full models, demonstrating practical applicability.

**Weaknesses:**

The paper does not clearly position itself in relation to previous robustness studies. For instance, Chhipa et al. (2024) [1] evaluated open-vocabulary detectors including OWL-ViT, YOLO-CLIP, and Grounding DINO under distribution shifts and common corruptions, reporting significant performance drops across models. Similarly, Liu et al. (2024) [2] introduced robustness benchmarks such as COCO-C and BDD100K-C in the broader object detection literature, showing that even high-mAP detectors can be fragile, while transformer-based architectures generally perform better under corruptions. It would strengthen the paper to connect its analysis with these prior works and clarify how its findings extend or differ from them.

Evaluation noise types are somewhat narrow, focusing primarily on pixelation, turbulence, and motion blur, with little exploration of other real-world distortions (e.g., rain, snow, fog).

[1] Chhipa, Prakash Chandra, et al. (2024) "Open-Vocabulary Object Detectors: Robustness Challenges Under Distribution Shifts." European Conference on Computer Vision.

[2] Liu, Jiawei, et al. "Benchmarking object detection robustness against real-world corruptions." International Journal of Computer Vision, 132 (10), 4398-4416. (2024)

**Questions:**

Please follow the strengths and weaknesses.
I am open to adjusting scores.

---

> ### Author Response · Authors · 2025-11-18
> **Rebuttal - 1**
>
> We are grateful to the reviewer for their positive assessment. We would like to use this opportunity to clarify the few points of ambiguity, which will be integrated into the final manuscript.
>
> ---
> ---
> ```
> W1. previous robustness studies.
> ```
> Thank you so much for the constructive reference. We will include the following elaboration in the final copy to differentiate our work from the existing work.
>
> **Core Key difference:**
> ```
> [1] Chhipa, Prakash Chandra, et al. (2024) "Open-Vocabulary Object Detectors: Robustness Challenges Under Distribution Shifts." European Conference on Computer Vision.
> ```
> The referred work [1] basically defines robustness as the distribution shift impact on models (via COCO-O, and COCO-DC), e.g. high-quality image of a painting, cartoon, or tattoo. While our work primarily deals with noises, which have a unique property of feature collapse (similar to COCO-C). Our work `(Fig. 2)`, including some recent studies [3, 4, 5, 6], has shown that noise is essentially feature/variance collapse, i.e., noise, despite what they look like in RGB space, is similar with respect to feature collapse. Establishing that every form of distribution shift (like painting, cartoon, or tattoo) as a feature collapse is beyond the scope of our work.
>
> ```
> [2] Liu, Jiawei, et al. "Benchmarking object detection robustness against real-world corruptions." International Journal of Computer Vision, 132 (10), 4398-4416. (2024)
> ```
> Similar differences with [2] exist: noises like illumination, black lines, and color do not necessarily mean feature collapse, and establishing it is beyond the scope of our work. Another key difference is that it's a ResNet-heavy (non-open vocabulary), they train (fine-tune) their models on COCO before analysis, while we are primarily an open vocabulary model with both zero-shot and finetuning-based robustness analysis.
>
> Overview of key differences in “empirical” insights of [1,2] include :
> | Category | [1] ECCV'24 | [2] IJCV'24   | Our (Robust Onion)
> | :-------: | :------- | :------- |  :------- |
> | **Type** | Zero-shot Benchmark on COCO Variants on **2 transformers** | Analysis (finetuned) on COCO Variants and BBDK100 for ResNet and **2 transformers**.  | Transformer-centric analysis of zero-shot & fine-tuning on COCO Variants (noise, LVIS, RefCOCO / RefCOCOg / RefCOCO+), ODinW-13, Flickr30k, Wider Face with **29 transformer variants**. |
> **Goal**  | Black-box evaluation (No unboxing) to determine which models are better under what conditions (without determining the underlying cause) | Non-open vocabulary ResNet analysis (No unboxing) focusing on training strategy | Investigation-centric (Answer “Why” / “Where” / ”How” there is a performance drop in open-vocabulary transformers).  Benchmark in setup `(Sec 3 & Fig. 5)`. Minimal focus on ResNets.
> **Takeaway [1]** | Grounding DINO exhibits the least performance drop compared to OWL-ViT, YOLO World on COCO-C | | We explain this via **performance linear relationship with robustness** (`Figure 5b)`), so “Grounding DINO (48.4) > YOLO World (39.3) > OWL-ViT (26.4)” implies robustness ordering as well.
> **Takeaway [2]** | | mAP and robustness are not linear.  | We explain this by indicating: **Similar backbones have similar robustness** (`Figure 6b)`), and DETR and Deformable DETR have the same transformer backbone.
>
> [3] LR0.FM: Low-Res Benchmark and Improving Robustness for Zero-Shot Classification in Foundation Models. ICLR 2025
> [4] Mint: A Simple Test-Time Adaptation of Vision-Language Models against Common Corruptions, Neurips 2025
> [5] Dive into the Resolution Augmentations and Metrics in Low Resolution Face Recognition: A Plain yet Effective New Baseline. AAAI 2023
> [6] Recognizability Embedding Enhancement for Very Low-Resolution Face Recognition and Quality Estimation, CVPR 2023.

---

> ### Author Response · Authors · 2025-11-18
> **Rebuttal - 2**
>
> ```
> W2. Evaluation on real-world distortions (e.g., rain, snow, fog).
> ```
>
> We are thankful for the constructive advice on making our work more generalizable. Our work (`Fig. 2`), including some recent studies [3, 4, 5, 6], has demonstrated that noise is essentially feature/variance collapse, despite its appearance in RGB space. That's the core of our analysis in `Figure 8`, explaining why similar depths have similar robustness.
>
> We have chosen **Turbulence** as a representative of the group exhibiting “Observable Collapse” (this includes Rain, Snow) and **Motion Blur** from the group exhibiting “Minimal Collapse” (`Fig. 2`). These groups correspond to real self-driving car datasets' noises as well, where “Observable Collapse” encompasses cloudy and overcast skies, and “Observable Collapse” encompasses Snow and Rain.
>
> One of the key takeaways of our work is:
> *“​The choice of backbones decides the robustness, while other bells and whistles, like additional modules (e.g. MMGDINO/GLEE decoder), and extensive pre-training (e.g. MM-GDINO pre-trained on 9 datasets, and GLEE on 18 via three stages of training etc.) play a minimal role.”* (`line 266`).
> We showed this analysis for **all the noises in this study, including rain and fog** `(supplementary Fig 28, values and shades of blue)`.
>
> **For Rain:**
> *Swin-T (MM-GDINO, GLIP) < Swin B (MM-GDINO) ~ Swin-L (MM-GDINO, GLIP, GLEE) <= EVA (GLEE)*
> Which is similar to Turbulence from the similar feature collapse group
> *Swin-T (MM-GDINO, GLIP) < Swin B (MM-GDINO) ~ Swin-L (MM-GDINO, GLIP, GLEE) ~ EVA (GLEE)*
>
> **For Snow:**
> *Swin-T (MM-GDINO, GLIP) < Swin B (MM-GDINO) ~ Swin-L (MM-GDINO, GLIP, GLEE) ~ EVA (GLEE)*
> Which is almost identical to the motion blur from the similar feature collapse group
> *Swin-T (MM-GDINO, GLIP) < Swin B (MM-GDINO) ~ Swin-L (MM-GDINO, GLIP, GLEE) ~ EVA (GLEE)*
>
> Different models have similar robustness because of similarity in backbones. And Swin-B and Swin-L performance are almost similar because of the depth of 24.
>
> Finally, we took a step further in synthetic simulation by validating our analysis on the **Wider Face dataset** in `Section 5 (LR-TK0++)`, demonstrating that our insights regarding backbone robustness can translate directly to improving performance on naturally occurring, uncontrolled noise in the wild.
>
> ---
> ---
> We would like to thank you once again for the feedback and will be happy to answer additional questions/concerns.
>
> ---
> ---
>
> [3] LR0.FM: Low-Res Benchmark and Improving Robustness for Zero-Shot Classification in Foundation Models. ICLR 2025
> [4] Mint: A Simple Test-Time Adaptation of Vision-Language Models against Common Corruptions, Neurips 2025
> [5] Dive into the Resolution Augmentations and Metrics in Low Resolution Face Recognition: A Plain yet Effective New Baseline. AAAI 2023
> [6] Recognizability Embedding Enhancement for Very Low-Resolution Face Recognition and Quality Estimation, CVPR 2023.

---

### Official Review · Reviewer_oUsL · 2025-11-01

**Soundness:** 3
**Presentation:** 3
**Contribution:** 3
**Rating:** 8
**Confidence:** 4

**Summary:**

This work evaluates the effect of noise on open-vocabulary object detectors, similar to ImageNet-C and CIFAR-C for classification. The authors analayze numerous detectors across many axes. Interesting findings include robustness being correlated to backbone, and that robustness is more correlated to images than corresponding annotations; while pre-traiining  and captions matter little. They propose a solution, training spatial tokens, on corrupted inputs to improve robustness.

**Strengths:**

* The analysis is thorough and interesting. I reallt enjoyed reading all of Section 4, and I found the key findings, such as sensitivity to backbone, to be very interesting.
* The presentation is mostly very good.
* There was a need for a such a study in the literature; object detectors are related to yet different from classifiers, and so one might see different behaviour.

**Weaknesses:**

* Although the paper is mostly well written, Section 5 became difficult to parse. For example in line 450 "an existing approach for low-resolution clas- sification (preserves zero-shot)"; what does preserves zero-shot mean here?
* It seems like the solution in Section 5 trains on corrputions. This seems to be  training on the test domain, in which case this ceases to properly test generalization.
* Some findings are not suprising; like larger objects being more robust. It's more difficult to corrupt the structure!

**Questions:**

* When visual backbones are shared, couldn't they have the same visual pre-training? So maybe it's not suprrising that they would be correlated in robustness.
* Will a benchmark be released?
* For classifiers there is a concept of expected calbiration error for measuring decreased confidence in ood examples. Is there a similar metric that could be shown here? It seems like this could be an good measure of 'graceful degradation'.

---

> ### Author Response · Authors · 2025-11-18
> **Rebuttal  - 1**
>
> We sincerely thank the reviewer for their positive assessment. We are grateful for the opportunity to clarify the few points of ambiguity, which will be integrated into the final manuscript.
>
> ---
> ---
> ```
> W1 What does preserves zero-shot mean here?
> ```
> We apologize for the lack of clarity. "Preserves zero-shot" means that the proposed lightweight adaptation methods (LR-TK0+ / LR-TK0++) do not fine-tune the original, frozen model weights, retaining (preserving) the clean HQ accuracy (zero-shot accuracy). The final copy will have 1 extra page that will reflect :
> - The core OV-OD model (GLIP-T) remains completely frozen.
> - Only a small set of newly inserted spatial tokens are trained.
> - Because the original model is unaltered, its learned ‘general-purpose knowledge’ and its ability to detect unseen classes (its "zero-shot" capability) are not overwritten or damaged. This avoids the "catastrophic forgetting" that would occur with standard full-model fine-tuning.
> ---
> ---
> ```
> W2. Section 5 trains on corrputions.
> ```
> As it is a critical point, we apologize if our setup was unclear. We have not trained on the test domain. The model is trained on synthetic pixelation and tested on **unseen real-world noise (Wider Face), which may or may not have pixelation**.  To further clarify, we:
> - *Training Domain*: Our models are trained on a small, 3,000-image subset of standard pretraining dataset Flickr30k, using synthetic pixelation. `(line 468)`
> - *Test Domains*: Evaluation in a zero-shot setting, on two completely separate domains: (1) the synthetic COCO (pixelation) benchmark and (2) the real-world, naturally-corrupted Wider Face dataset (we speculate it might have pixelation). `(line 469)`
> - *Generalization*: The key result is that training on synthetic, out-of-domain noise (Flickr30k) generalizes to improve robustness on both unseen synthetic domains (COCO) and, more importantly, on unseen real-world noise (Wider Face) `(Tab 2)`.
> - This validates that the models like GLIP (have 3 transformers and multiple architectural components) can improve robustness just by tweaking visual backbones, especially at the shallow layers `(Sec 4.1 and 4.2)`.
> ---
> ---
> ```
> W3. Larger objects being more robust.
> ```
> We appreciate the reviewer carefully parsing all our analysis. We agree that this specific finding is unsurprising and intuitive; at the same time, it serves as a crucial explanation of “why a dataset like  ODinW-13 has far superior robustness compared to other datasets like COCO / LVIS” `(Line 354)`. ODinW-13 has only ~10% of very small objects (<= 32 × 32) compared to ~42% in COCO and ~58% in LVIS.
>
> ---
> ---
> ```
> Q1. When visual backbones are shared, couldn't they have the same visual pre-training?
> ```
> We appreciate the reviewer for raising this excellent question, as it's a core finding of our analysis. The reviewer is correct that shared backbones are used, but they are not identically trained (different from pre-training). For example, after similar ImageNet (1 Million - 15 Million) pre-training of Swin Transformers, these models undergo different stages of training **~400 Million - 10 Billion image-text pairs in different stages** (distillation, pretraining, fine-tuning). Models like GLEE, MM-GDINO, and GLIP use vastly different pre-training datasets and strategies (e.g., GLEE on 18 datasets in 3 stages, MM-GDINO on 9, GLIP via distillation).
>
> Our observation `(Section 4.1)` shows that despite these different pre-training schemes, similar backbones have similar robustness. This strongly implies that the "bells and whistles of pretraining or overall architecture" play a fairly minimal role. The robustness is not correlated because of shared pre-training (which is diverse), but because the backbone architecture itself is the dominant factor, overriding the influence of pre-training data.
>
> ---
> ---
> ```
> Q2. Will a benchmark be released?
> ```
> Yes. We will publicly release all code required to generate noise, evaluation scores, and LR-TK0+ and LR-TK0++ implementations, facilitating any researcher to apply our framework to their own models or datasets.

---

> ### Author Response · Authors · 2025-11-18
> **Rebuttal - 2**
>
> ```
> Q3. Expected calibration error
> ```
> This is a fantastic suggestion. While we focused on relative mAP (a standard for detection), analyzing model confidence is a critical next step.
>
> - Expected Calibration Error (ECE) for detectors is non-trivial, as these are open-vocabulary detectors with no fixed classifier as such. The classification logit is the alignment of vision features with text features. Though an approximation via localization (bbox regression) and classification confidence can be drawn.
> - We strongly believe vision backbone features get impacted by noise the most. Any metric that deals with subtle feature changes (popularly known as feature collapse) can be a good metric for determining the “graceful degradation.”
> - Recent work [1] proposed using intra-feature variance as a measure of feature collapse or a metric of noise degradation.
> - We appreciate this valuable direction for future work and will add this as a key avenue for exploration in our conclusion.
>
> ---
> ---
> We would like to thank you once again for the feedback, and we will be happy to answer additional questions/concerns.
>
> ---
> ---
>
>
> [1] Mint: A Simple Test-Time Adaptation of Vision-Language Models against Common Corruptions, Neurips 2025

---

> ### Comment · Reviewer_oUsL · 2025-11-24
> **Thank you**
>
> I thank the authors for their thoughtful response. Overall, I am satisfied and will keep my positive rating of 8. I have one remaining comment:
>
> **Our observation (Section 4.1) shows that despite these different pre-training schemes, similar backbones have similar robustness. This strongly implies that the "bells and whistles of pretraining or overall architecture" play a fairly minimal role. The robustness is not correlated because of shared pre-training (which is diverse), but because the backbone architecture itself is the dominant factor, overriding the influence of pre-training data.**
>
> It is my understanding that initial stages of pre-training on eg. ImageNet has an outside effect on the function learned by the model. So, even though further training can be quite different, I am not certain this claim is fully sound. I recomment softening it a bit.

---

> ### Author Response · Authors · 2025-11-25
> **Thanking Reviewer**
>
> Thank you for acknowledging our rebuttal, really appreciate it. Accepting the above recommendation, we will rephrase the `[line 266] (final revised copy)` to a softer tone while keeping the door to the future research open:
>
> *"Models with similar backbones tend to exhibit similar robustness, suggesting that the backbone architecture has a substantial influence on robustness, while the additional architectural modules (e.g. MM- GDINO / GLEE decoder, neck / FPN network, classifiers, etc) and training strategies (e.g. MM-GDINO pre-trained on 9 datasets, and GLEE on 18 via three stages of training, and GLIP via two stages etc.) play a minimal role. While the role of initial pre-training of visual backbones (e.g., on ImageNet, Conceptual Captions [2])  on the final robustness cannot be completely ruled out, our results indicate a stronger architectural effect rather than that of pre-training."*
>
> [2]  Piyush Sharma, Nan Ding, Sebastian Goodman, and Radu Soricut. Conceptual captions: A cleaned, hypernymed, image alt-text dataset for automatic image captioning. In ACL, 2018

---

### Author Response · Authors · 2025-11-18
**Thanking all the reviewers**

We appreciate every reviewer taking their valuable time to review our paper and ask crucial questions, helping us to highlight more clarity in our work. Our work is an example of **"interpretability and explainable AI"** work, with the goal to analyze the influence of noises on OV-ODs (`line 79`) & narrow down the bottlenecks, and *not necessarily to propose a solution* for fixing the models (`line 95`). In simple words, our work intends to open up the black box of (OV-ODs) to see how and where noise impacts these models.

We appreciate all the reviewers for acknowledging some of the clear goals of our work :
 - Through detailed analysis with some interesting insights (`oUsL`,  `x5K3`,  `gY5W`, `cRjy`)
 - Analysis of Open Vocab Object Detectors (VLMs) as opposed to traditional classifier-based analysis. (`oUsL`)
 - Peeling layers of complex models as opposed to black box benchmarking (`gY5W`)
- Relevance of our insights in designing practical orbust models (`x5K3`, `gY5W`, `cRjy`)
- Practical relevance of LR-TK0+/TK0++ (`x5K3`, `gY5W`)


While addressing each reviewer's concern, the rebuttal broadly highlights the following:
- `Section 5 (LrTK0+ & LrTK0++)` is a **"validation of our analysis"** (section is renamed to reflect it), and **not a proposed technique** for explicitly improving robustness. (`gY5W, cRjy`). Apologies for the confusion
- Clear distinction between our work (answering “Why” / “Where” / ”How” performance drops in OV-ODs) vs. existing works. (`x5K3, gY5W`)
- Clarification on comparison with standard methods (e.g., augmentation, in `Tab 1`).  (`gY5W, cRjy`)
- Highlight of more analysis done for more noises like Fog, Rain, Snow in Supplementary (`Fig. 28`) (`x5K3, gY5W`).

---

### Author Response · Authors · 2025-12-02
**[ Post Rebuttal ] Summarizing the Entire Rebuttal Process**

### **Short Answer**
We thank the Reviewers and the Area Chair for their effort and thoughtful feedback. All reviewers agreed on the clarity, depth, and value of our **interpretability and explainable AI** empirical analysis, comprising of thorough analysis, comprehensive evaluation, never-before-seen inferences, and actionable insights for Open-Vocabulary Object Detectors against real-world noises. We fully addressed all comments through detailed rebuttals, clarifying that our primary contribution is a detailed analysis of **opening the black box of object detectors** under noises with a simple strategy for validating our insights and not a proposed robustness method. These revisions will be re-emphasized in the revised version (extra 10th page).


`Reviewer oUsL` maintained a **positive score of 8**. `Reviewers x5K3 and gY5W` explicitly stated they would raise their scores **(6 →  ≥ 8 and 4 →  ≥6)** after satisfactory rebuttals, which we have fully addressed. `Reviewer cRjy’s` concerns stemmed from a misunderstanding of the paper type, which we have accordingly clarified as our work being an empirical *interpretability and explainable AI* analysis.

Our conservative projection of final scores at least could have been `8,8,6,4` with 2 → 4 low on confidence (3). Given that the majority of reviewers leaned towards positive reviews, with two explicitly moving to higher positive scores, we hope the final decision reflects the clarity and completeness of our revisions.

---
---
### **Long Detailed Answer**
Across all reviewers, the common consensus supports our overall goals of  **interpretability and explainable AI** as:
- *“rich”, “methodologically”, “clear”, “thorough and interesting”* analysis (`cRjy, gY5W, x5K3, oUsL`).
- *“Comprehensive evaluation across multiple models and datasets”* (`cRjy, x5K3`)
- *“Diverse visual distortions”*  (`cRjy,x5K3`)
- Delivering *“some important”* and *“actionable”* insights regarding robustness `(gY5W, x5K3)`.
- Analysis beyond  *“black box”* evaluation. (`gY5W`)


We have carefully addressed every comment raised during the rebuttal phase through detailed, point-by-point rebuttals and multiple clarifications, mostly requiring rephrasing of the introduction as empirical analysis rather than a proposal for robustness. The issues raised were mostly around this confusion rather than technical shortcomings, which will be fully incorporated into the final revised copy with an extra 10th page. Summarizing comments from each reviewer,

---
`Reviewer oUsL` acknowledged that all clarifications were satisfactorily addressed, with a **high positive score of 8**, with minor rephrasing in the final copy.

---
`Reviewer x5K3` explicitly stated that they would **raise their scores from 6** upon satisfactory rebuttals (`I am open to adjusting scores`), which we have addressed by highlighting:
- How our in-depth analysis is fundamentally different from previous results-driven benchmarks and how we can even explain the empirical findings of the referenced works (w1).
- Reference to supplementary highlighting analysis on additional noises like rain, snow, fog (w2).

---
`Reviewer gY5W` also explicitly stated that they would **raise their scores from 4** upon satisfactory rebuttals (`I can change the score based on rebuttal's response.`), which we have addressed via:
- How our work is an empirical study under interpretability and explainable AI, with a simple strategy for validating our insights and not a proposed robustness method (w1, w3).
- How our in-depth analysis is fundamentally different from previous results-driven benchmarks and how we can even explain the empirical findings of the referenced works (w2).
- References in the main paper supporting the claim *“visual backbone determines the robustness”* (w4)
- Reference to supplementary highlighting analysis on additional noises like rain, snow, fog (w5).

---
`Reviewer cRjy` reasoning for rating of 2 stemmed from a fundamental misunderstanding of the nature of our work, specifically, confusion about whether we were proposing a new method or conducting an interpretability-driven empirical analysis (s1-4, w3). We fully clarified this distinction, reinforced our intended contribution as
- How our work is an empirical study under interpretability and explainable AI, with a simple strategy for validating our insights and not a proposed robustness method (w1, w3, w4, w5, q1, q4).
- Expansion of motivation for studying robustness against real-world noises (w2, q2)
- References in the main paper highlighting all design proposals for improving robustness of the model (w3, q3)

---
The final copy will reiterate and emphasize the nature of work as a deep analysis far beyond the black box evaluation. Given that the majority of reviewers were leaning towards positive scores, with two explicitly moving to higher positive scores, we hope the final decision will reflect the clarity and completeness of our revisions.

---

### Meta-Review · Area_Chair_JxSM · 2026-01-05

**Summary:**

This paper presents *Robust Onion*, a large-scale empirical analysis of open-vocabulary object detectors under synthetic visual degradations. The authors evaluate multiple OV-OD architectures, datasets, and noise types to study which components most strongly influence robustness. The paper reports several empirical observations, including that robustness is largely driven by the vision backbone, that annotation type and language prompts have limited impact, and that benchmarks such as ODinW-13 may overestimate robustness due to dataset bias.

Reviewers generally agree that the paper is thorough, clearly motivated, and experimentally extensive. However, there is also a consistent concern across multiple reviews that the work remains largely descriptive, with limited conceptual integration across analyses, and that the link between empirical findings and proposed design implications is not sufficiently rigorous or clearly validated.

**Reviewer Concerns:**

Several reviewers raised concerns about the positioning and contribution of the paper (x5K3, gY5W, cRjy). While the rebuttal clarifies that the work is intended as an interpretability-focused empirical study rather than a new robustness method, this clarification does not fully resolve the core issues raised in the reviews.

**Unclear validation of design implications.**
 The paper discusses several design implications derived from the empirical analysis, but their validation is limited. In particular, the role of Section 5 is not clearly positioned with respect to the earlier analysis, blurring the distinction between analysis-driven validation and method-level contribution. As a result, it remains unclear to what extent the proposed intervention supports the analytical claims (gY5W).

**Primarily descriptive, correlation-based analysis.**
 While the paper reports a large number of empirical observations, several reviewers noted that the analysis remains largely descriptive and correlation-based. Key conclusions, such as the dominant role of the backbone or the limited impact of language, are plausible and empirically supported, but are not accompanied by deeper explanation or justification of why these effects arise (x5K3, gY5W).

**Limited disentanglement of interacting factors.**
 The factors examined in the paper are mostly analyzed in isolation. Reviewers expressed concerns that potentially confounding factors are not sufficiently disentangled, making it difficult to determine whether the reported trends would hold under more careful or joint consideration of multiple variables (x5K3, cRjy).

Reviewers also raised concerns regarding the range of noise types considered and the paper’s positioning with respect to prior robustness studies. The rebuttal provides additional clarification and discussion on these points, which improves the clarity of scope and context. However, these clarifications do not address the more fundamental concerns about the depth, integration, and validation of the analysis discussed above.

**Reviewer Scores:**

Reviewer oUsL remained strongly positive, valuing the scale and clarity of the empirical study. Reviewers x5K3 and gY5W indicated that their scores could potentially change with clarification, but there is no clear evidence that they would necessarily increase their scores post-rebuttal. Reviewer cRjy remained unconvinced, primarily due to concerns about limited conceptual depth and rigor.

Overall, despite improved clarity after rebuttal, the balance between strengths and weaknesses remains largely unchanged.

---

### Decision · Program_Chairs · 2026-01-26

Reject